# Automated selection of mid-height intervertebral disc slice in traverse lumbar spine MRI using a combination of deep learning feature and machine learning classifier

Friska Natalia[1], Julio Christian Young[1], Nunik Afriliana[1], Hira Meidia[1], Reyhan Eddy Yunus[2], Sud Sudirman[3] *

1 Faculty of Engineering and Informatics, Universitas Multimedia Nusantara, Serpong, Indonesia, 2 Dr. Cipto Mangunkusumo Hospital, Jakarta, Indonesia, 3 School of Computer Science and Mathematics, Liverpool John Moores University, Liverpool, United Kingdom

* s.sudirman@ljmu.ac.uk

**Data Availability Statement:** We have made our dataset and source code published and open to the public. They are on Mendeley Data. URL: https://

## Abstract

Abnormalities and defects that can cause lumbar spinal stenosis often occur in the Intervertebral Disc (IVD) of the patient's lumbar spine. Their automatic detection and classification require an application of an image analysis algorithm on suitable images, such as mid-sagittal images or traverse mid-height intervertebral disc slices, as inputs. Hence the process of selecting and separating these images from other medical images in the patient's set of scans is necessary. However, the technological progress in making this process automated is still lagging behind other areas in medical image classification research. In this paper, we report the result of our investigation on the suitability and performance of different approaches of machine learning to automatically select the best traverse plane that cuts closest to the half-height of an IVD from a database of lumbar spine MRI images. This study considers images features extracted using eleven different pre-trained Deep Convolution Neural Network (DCNN) models. We investigate the effectiveness of three dimensionality-reduction techniques and three feature-selection techniques on the classification performance. We also investigate the performance of five different Machine Learning (ML) algorithms and three Fully Connected (FC) neural network learning optimizers which are used to train an image classifier with hyperparameter optimization using a wide range of hyperparameter options and values. The different combinations of methods are tested on a publicly available lumbar spine MRI dataset consisting of MRI studies of 515 patients with symptomatic back pain. Our experiment shows that applying the Support Vector Machine algorithm with a short Gaussian kernel on full-length image features extracted using a pre-trained DenseNet201 model is the best approach to use. This approach gives the minimum per-class classification performance of around 0.88 when measured using the precision and recall metrics. The median performance measured using the precision metric ranges from 0.95 to 0.99 whereas that using the recall metric ranges from 0.93 to 1.0. When only

data.mendeley.com/datasets/ggjtzh452d/1 DOI: 10.17632/ggjtzh452d.1 Further instructions and information can be found in the paper's Supporting information files.

**Funding:** Grant Holder: FN Grant Number: 9/E1/KPT/2020 Funder: The Indonesian Ministry of Research, Technology and Higher Education. Funder URL: https://www.ristekbrin.go.id/ The funder had no role in study design, data collection and analysis, decision to publish, or preparation of the manuscript.

**Competing interests:** The authors have declared that no competing interests exist.

considering the L3/L4, L4/L5, and L5/S1 classes, the minimum F1-Scores range between 0.93 to 0.95, whereas the median F1-Scores range between 0.97 to 0.99.

## 1. Introduction

The success of many modern therapeutics for an illness relies on speedy and accurate diagnoses of the illness. And obtaining speedy and accurate diagnoses of illnesses is a fundamental challenge for global healthcare systems. This is the reason why computer-aided diagnosis (CAD) is seen as a potential solution to overcome this challenge. A CAD system can help doctors understand the cause of an illness better by automating some steps in the diagnosis process. In a CAD system that uses medical images, the system applies image analysis algorithms to different types or modalities of medical imaging, such as Magnetic Resonance Imaging (MRI), of the patient [1–3]. In the case of MRI, for example, a CAD system might use the two modalities of MRI, namely the T1-weighted and T2-weighted MRI, which can differently highlight various types of tissues based on their fat and water composition. An example of a T1-weighted and a T2-weighted traverse MRI images of the L3/L4 Intervertebral Disc (IVD) of the same patient are shown in Fig 1. The algorithms may also require images with specific properties and criteria as inputs. Some algorithms require mid-sagittal MRI images as inputs [4–6] whereas some others require traverse images taken at certain locations as inputs [7–10]. When these algorithms were proposed in the literature, it is often assumed that a selection process has been carried out beforehand that identifies appropriate and suitable images as their input. In practice, however, these selection processes are not straightforward since a patient's data repository contains more than just these specific images, hence the process to select the

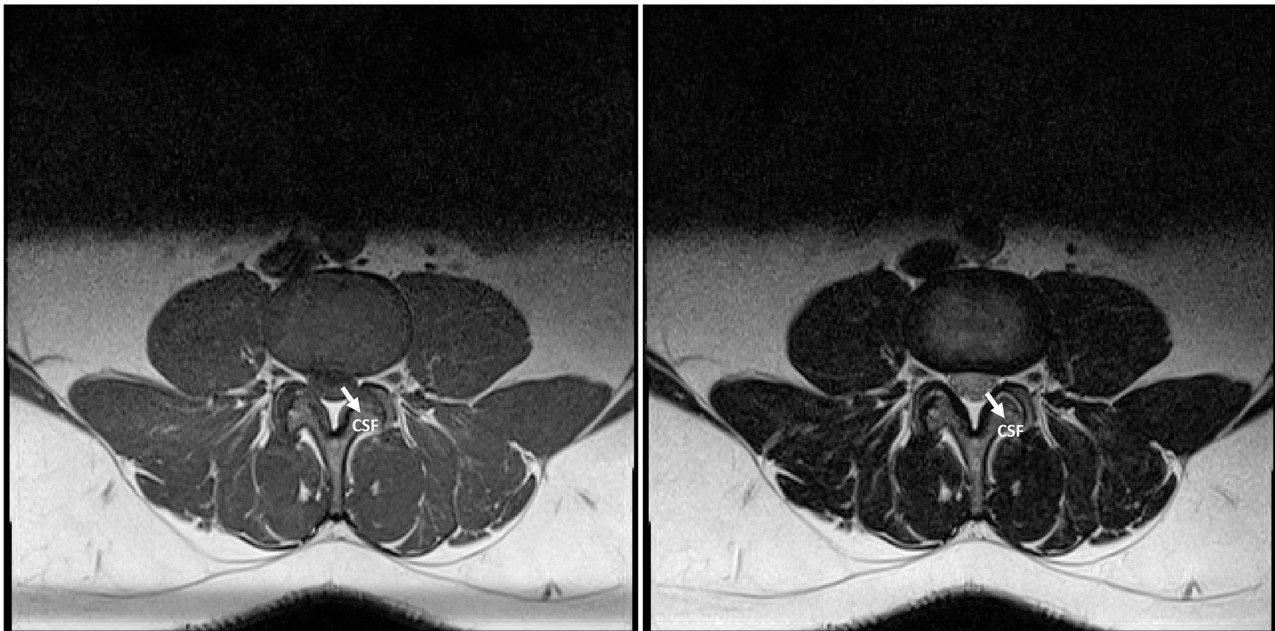

**Fig 1. A T1-weighted (left) and a T2-weighted (right) traverse MRI images of the L3/L4 Intervertebral Disc of a patient are shown.** One marked difference in the two images is the cerebrospinal fluid (CSF) in the spinal canal that appears black on the T1-weighted image but as a brighter region on the T2-weighted image because of its low fat contents.

suitable images is often done manually. Therefore, to make the CAD system more automated, this selection process also needs to be automated.

Our review of the literature discovers that the classification of brain MRI images, and more particularly the selection of mid-sagittal plane in brain MRI, has been a popular research topic in the last three decades [11] due to its uses in brain image processing such as spatial normalization, anatomical standardization, and image registration. However, the topic of plane selection in spinal images has not taken up much researchers' attention despite the need for such technology. To the best of our knowledge, no method in the literature has been proposed to select the best traverse plane that cuts closest to the half-height of an IVD in a lumbar spine MRI. These images are very useful for detecting abnormalities in lumbar IVDs including those resulted from Lumbar Spine Stenosis (LSS), a condition that causes low back pain because of the pressures exerted on the spinal nerve [12]. Most LSS occurs in the last three lumbar IVDs namely the L3/L4, L4/L5, and L5/S1 IVDs due to the heavier weight they have to support in comparison to other IVDs and MRI is the most commonly used imaging technology for diagnosing LSS due to its high soft-tissue resolution [12]. When attempting to diagnose LSS using MRI, a neuroradiologist almost always starts his or her inspection of those three IVDs in the mid-sagittal view (as illustrated in Fig 2) since it can provide a general overview of the lumbar spine's condition. This is also reflected in the popularity of image analysis methods in the literature that use sagittal MRI images to detect LSS [6, 13–16]. However, a more accurate assessment of the actual location and the extent of the LSS can only be obtained through inspection of the suspected IVD in traverse view (as illustrated in Fig 3) [17, 18]. Traverse images taken from planes that cut closest to the half-height of the L3/L4, L4/L5, and L5/S1 IVDs are generally considered as the best images to use when the neuroradiologist inspects the disc because

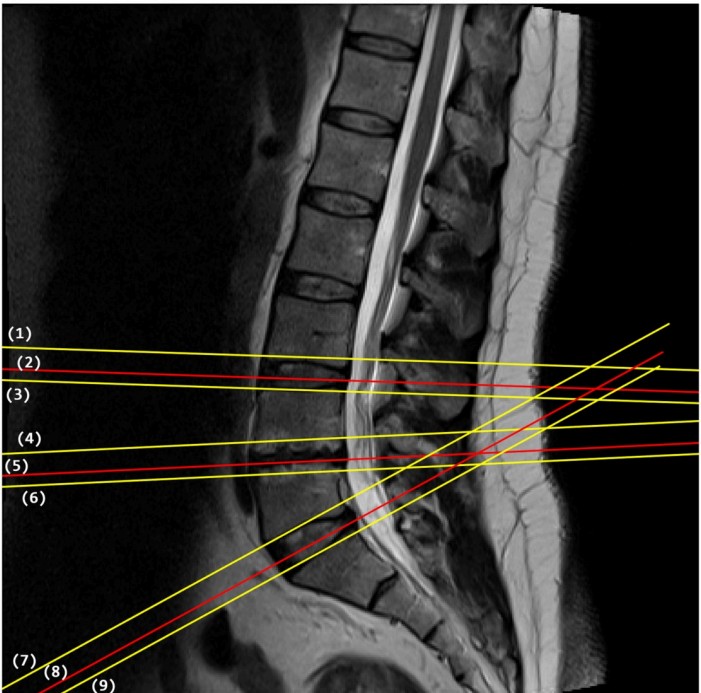

**Fig 2. A mid-sagittal view of a lumbar spine MRI showing the intersection lines between the sagittal plane and the traverse planes that are shown in Fig 3.** The lines marked in red are the intersection lines of traverse planes that cut closest to the half-height of an IVD.

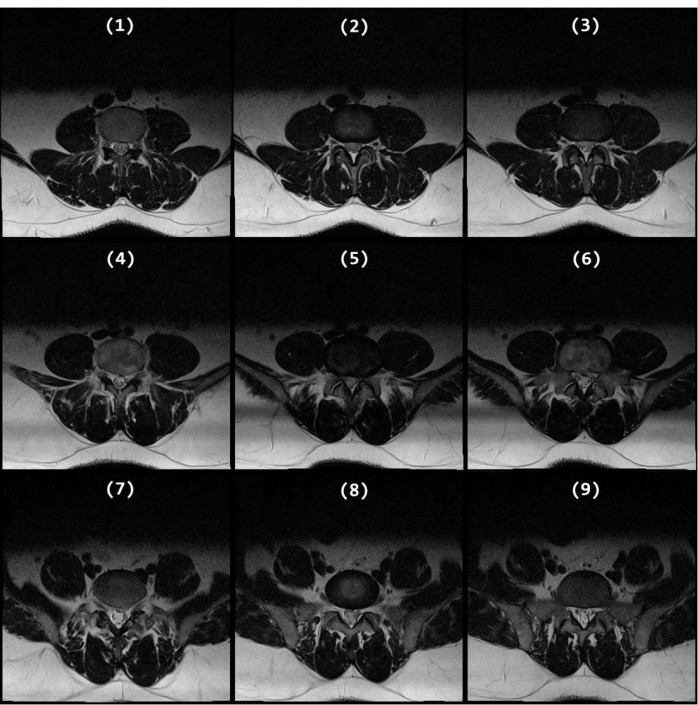

**Fig 3. An example of nine traverse images of a lumbar spine.** Image 2, 5, and 8 are from the planes that cut closest to the half-height of L3/L4, L4/L5, and L5/S1 IVD, respectively.

they contain the best information that reflects the condition of the IVDs. Fig 2 shows the intersection lines between the mid-sagittal plane and each of the nine traverse planes. The lines marked in red are the intersection lines, which according to our expert's view, of traverse planes that cut closest to the half-height of an IVD.

The task of selecting these traverse images falls into the category of image classification and can be solved using machine learning (ML). A typical approach in image classification using machine learning involves two stages, with the first being the extraction of relevant information from the images via the calculation of low-level handcrafted features [19–21]. This is then followed by a classification of the calculated features using trainable ML classifiers. Despite the success of this approach, it has a significant drawback when used in a wider image classification problem since the features are often task dependent. In other words, the handcrafted image features that are optimized for a particular task often perform poorly when used in a different task, and the accuracy of the classification is very dependent on the design of these features. Deep Convolutional Neural Network (DCNN) was proposed to overcome the problems associated with the traditional approach of image classification by allowing automatic learning of such features through forward and backward propagation of information in a series of convolutional and non-linear neural network layers [22, 23].

DCNN is one of several types of deep neural networks that gain popularity in recent years to solve many artificial intelligence problems. Different types of deep neural networks have significantly different architectures and are designed to solve different types of problems. DCNNs are typically used for image classification. Recurrent Neural Networks, such as Long Short-Term Memory [24], are used to recognize patterns in sequences of data such as time-series data, speech, and texts. There are also Fully Convolutional Neural Networks, such as U-net [25] and SegNet [26] that are used mainly for semantic image segmentation. Some DCNNs

have also been modified to become Region-based CNNs [27, 28] to detect and recognize multiple objects within an image.

Training DCNN models take a long time hence there exist several pre-trained DCNN models that are readily usable for image classification [29, 30]. Many of the most popular pre-trained DCNN models were developed using real (i.e., non-synthesized) photographic (i.e., non-medical) images from the ImageNet database [31] and the original task is to detect the types of objects that are typically present in photographs such as cars, fruits, animals, and so on. However, despite being extracted using a model trained using photographic images, these learnable features are sufficiently general that they can be used in many other types of image classification tasks, including medical image classification, through a method called Transfer Learning [32, 33], which process is elucidated in Fig 4. This method is performed by replacing the Fully Connected (FC) Neural Network classification layers of the DCNN with new ones before retraining them using the images from the new target dataset.

The classification of medical images has unique challenges compared to the classification of other more general images. Firstly, they have relatively high intra-class variation and inter-class similarity compared to other types of images [34] and secondly, the size of the medical image dataset is considerably smaller than datasets of other types of images. The latter is particularly problematic in the application of any machine learning methodology, including DCNN, because it violates the assumption that the number of samples is greater than the number of features. One of the possible solutions to solve this is through the application of Dimensionality Reduction (DR) or Feature Selection (FS) techniques to transform the data from a high-dimensional space into a lower-dimensional space while retaining much of the useful properties of the original data.

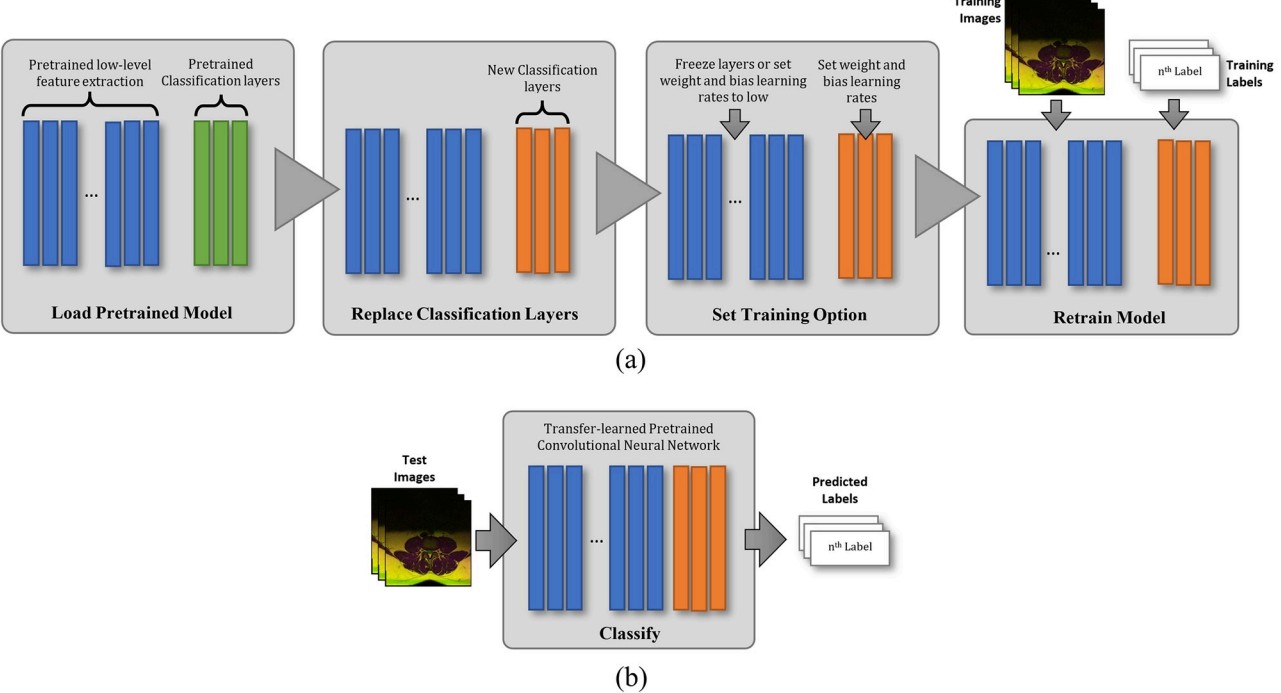

**Fig 4. A flowchart describing the traditional Transfer Learning approach of using Deep Convolutional Neural Network for medical image classification, where a) depicts the training process and b) depicts the inference step.**

Based on the above argument, we believe that both a) the lack of directly relevant methods proposed in the literature that selects the best traverse plane that cuts closest to the half-height of an IVD in a lumbar spine MRI and b) the wide range of potentially suitable DR or FS methods and image classification methods, provide the rationale and urgency for this study. The aim of this study is to find the best method to select the best traverse plane that cuts closest to the half-height of an IVD in a lumbar spine MRI by studying and comparing the different combinations of machine learning methods and approaches. We report the result of our investigation on the suitability and performance of different approaches of machine learning in solving the aforementioned medical image classification challenge. The contributions of this work are summarized as follows:

a) Investigated the classification performance using image features calculated using eleven different pre-trained DCNN models.

b) Investigated the effect of three dimensionality-reduction techniques and three feature-selection techniques on the classification performance.

c) Investigated the performance of five different ML algorithms and three FC learning optimizers which are trained with hyperparameter optimization using a wide range of hyperparameter options and values.

The organization of this paper is as follows. Section 2 describes the dataset used in the research and the proposed method. The experimental results, analysis, and discussion are discussed in detail in Section 3. We then provide the conclusion of our findings in the last section of the paper.

## 2. Material and method

We can confirm that all procedures performed in this study are in accordance with the ethical standards of both the United Kingdom and the Kingdom of Jordan and comply with the 1964 Helsinki declaration and its later amendments. The approval was granted by the Medical Ethical Committee of Irbid Speciality Hospital in Jordan where the original MRI dataset was procured. The data were analyzed anonymously.

The material used in this research is taken from our Lumbar Spine MRI Dataset which is available publicly [9, 35]. This dataset contains anonymized clinical MRI studies of 515 patients with symptomatic back pains. The dataset consists of 48,345 T1-weighted and T2-weighted traverse and sagittal images of the patients' lumbar spine in the Digital Imaging and Communications in Medicine (DICOM) format. The images were taken using a 1.5-Tesla Siemens Magnetom Essenza MRI scanner. Most of the images were taken when the patients were in the Head-First-Supine position, though a few were taken when they were in the Feet-First-Supine position. The duration of each patient study ranges between 15 to 45 minutes with time gaps between taking the T1- and T2-weighted scans ranging between 1 to 9 minutes. The patient might have made some movements between the T1 and T2 recordings, which suggests that corresponding T1- and T2- slices may not necessarily align and may require an application of an image registration algorithm to align them. The scanning sequence used in all scans is Spin Echo (SE), which is produced by pairs of radiofrequency pulses, with segmented k-space (SK), spoiled (SP), and oversampling phase (OSP) sequence variant. Fat-Sat pulses were applied just before the start of each imaging sequence to saturate the signal from fat matters to make it appear distinct to water. The range of acquisition parameter values used during traverse MRI scans is provided in Table 1.

**Table 1. The range of acquisition parameter values used during traverse MRI scans.**

| Sequence Types | T1-weighted | T2-weighted |
|---|---|---|
| Number of Echoes (ETL) | 3 | 9 to 16 |
| Repetition Time (milliseconds) | 385 to 953 | 1900 to 5000 |
| Echo Time (milliseconds) | 11.0 | 84.0 to 96.0 |
| Slice Thickness (mm) | 4.0 | 3.0 to 5.0 |
| Spacing Between Slices (mm) | 4.4 | 3.3 to 6.5 |
| Field of View (mm) | 220 | 220 |
| Matrix (Freq. x Phase) | 100% | 100% |
| Imaging Frequency (MHz) | 63.7 | 63.7 |
| Number of Phase Encoding Steps | 295 to 336 | 272 to 360 |
| Scanning Sequence | SE | SE |
| Sequence Variant | SK\SP\OSP | SK\SP\OSP |
| Scan Options | Fat-Sat | Fat-Sat |
| Number of Averages | 2 or 3 | 1 or 2 |
| Echo Train Length | 3 | 9, 13, 15, or 16 |
| Percent Sampling | 65 to 75 | 70 to 78 |
| Percent Phase Field of View | 96.9 to 100 | 90.6 to 100 |
| Pixel Bandwidth | 150 or 205 | 165, 190 or 225 |
| Flip Angle | 150 | 150 |

From the 48,345 images in the dataset, some 17,872 traverse images were taken. These traverse images cut across the lowest three vertebrae, the first sacrum, and the lowest three IVDs including the one between the last vertebrae and the sacrum. An example of the slicing position of these traverse images is shown in Fig 2. These 17,872 traverse images are made up of 8,936 pairs of T1-weighted and T2- weighted images. They are composed of 515 pairs that cut halfway across the height of L3/L4 IVD, 515 pairs that cut halfway across the height of L4/L5 IVD, 515 pairs that cut halfway across the height of L5/S1 IVD, and the other 7,391 pairs that do not cut halfway across the height of any IVDs. The categorization of the images is made by an expert radiologist by viewing the images using DICOM viewer software and manually identifying the three mid-height slices. We consider the radiologist's decision as the ground truth, which the results of automatic classification will be compared against.

From each pair of T1-weighted and T2-weighted images, a 3-channel composite image is created resulting in 8,936 composite images. The first channel of the composite image is constructed from the T1-weighted image, the second channel is constructed from the image-registered T2-weighted image, and the last channel is constructed from the Manhattan distance of the two. The image used to construct the second channel is obtained by performing image registration on the T2-weighted image to its T1-weighted counterpart to ensure that every pixel at the same location in both images corresponds to the same voxel in an organ or tissue. This is performed by finding the minimum difference between the fixed T1-weighted image and a set of transformed T2-weighted images calculated over a search space of affine transforms. Mathematically, the process can be described as follows: Let $I_R(v)$ be the reference 2D image and $I_T(v)$ be the to-be-transformed 2D image, where $v = [x, y, z]^T$ is a real-valued voxel location. The voxel location $v$ is defined on the continuous domains $V_R$ and $V_T$, that corresponds to each pixel in $I_R$ and $I_T$, respectively. Note that in our case, $I_R$ and $I_T$ are the T1-weighted image and the T2-weighted image, respectively. The image registration that we employ in this method is a process that seeks a set of transformation parameters $\hat{\mu}$ from all sets of

transformation parameters $\mu$ that minimizes the image discrepancy function $S$

$$\hat{\mu} = \arg\min_{\mu} S(I_R, I_T \circ g(v|\mu)) \tag{1}$$

We calculate $S$ using Matte's mutual information metric described in [36] over a search space in $\mu$ domain. The search process uses an iterative process called the Evolutionary Algorithm that perturbs, or mutates, the parameters from the last iteration. If the new perturbed parameters yield a better result than the last iteration, then more perturbation is applied to the parameters in the next iteration, otherwise a less aggressive perturbation is applied. The search process is optimized using the (1+1)-Evolutionary Strategy [37] which locally adjusts the search direction and step size and provides a mechanism to step out of non-optimal local minima. The search is carried out up to 300 iterations with a parameter growth factor of 1.05. A sequence of parametric bias field estimation and correction method, called PABIC [37], is applied to counter the effect of low-frequency inhomogeneity field and high-frequency noise on both T1 and T2 modalities.

Out of the 8,936 attempts to register the T1 and T2 images, 25 failed because the algorithm is unable to converge. This could be because the patient's position and orientation when the two scans were recorded differ significantly. In this case, the images were removed from the dataset resulting in 8,910 composite images. We show in Fig 5, two example cases where the image registration process succeeded (left column) and failed (right column).

Each composite image is labeled according to which group of images they are taken. They are labeled as *best_d3* for those that cut halfway across the height of L3/L4 IVD, or as *best_d4* for those that cut halfway across the height of L4/L5 IVD, or as *best_d5* for those that cut halfway across the height of L5/S1 IVD, or as *other_slices* otherwise. Since we have a class imbalance problem, we augment the dataset by randomly sampling the *other_slices* class and oversampling the other classes. The class population sizes before and after augmentation are shown in Table 2. The dataset is then split into two mutually exclusive subsets namely the training set and the test set with an 80:20 ratio, respectively. The training set is used to develop machine learning models for the image classification whereas the test set is used to measure the classification performance of the developed machine learning models.

The methodology that we adopt to solve the research challenge is finding the best combination of image features, feature dimension reduction or feature selection method (if applicable), and machine learning classifier or neural network from a comprehensive set of method combinations. An overview of the method used in this study is shown as a flowchart in Fig 6. It starts with an image-features extraction process using a pre-trained DCNN model. The image features are extracted by taking the outputs of the last feature extraction layer just before the classification layers. In practice, this is obtained by removing the classification layers before recording the output signals of the DCNN for each input image. Since the feature dimension is high we can reduce it by applying a DR or an FS technique. Both techniques reduce the feature's length while retaining much of the useful properties of the full-length feature. The difference being, while an FS technique only selects a subset of the dimension directly from the entire dimension set, a DR technique applies a transform function to the feature vector prior to selecting the subset. In this study, we investigated the applicability of three popular DR methods namely the Principal Component Analysis (PCA), Independent Component Analysis (ICA) [38], and Factor Analysis (FA) [39]. PCA uses linear transformation functions to separate the data into their major components by projecting them onto a set of feature subspaces that maximizes the components' variance. Rather differently from PCA, the ICA and FA methods use statistical methods to transform the data. ICA transforms the data into independent non-Gaussian components by maximizing the statistical independence of the estimated

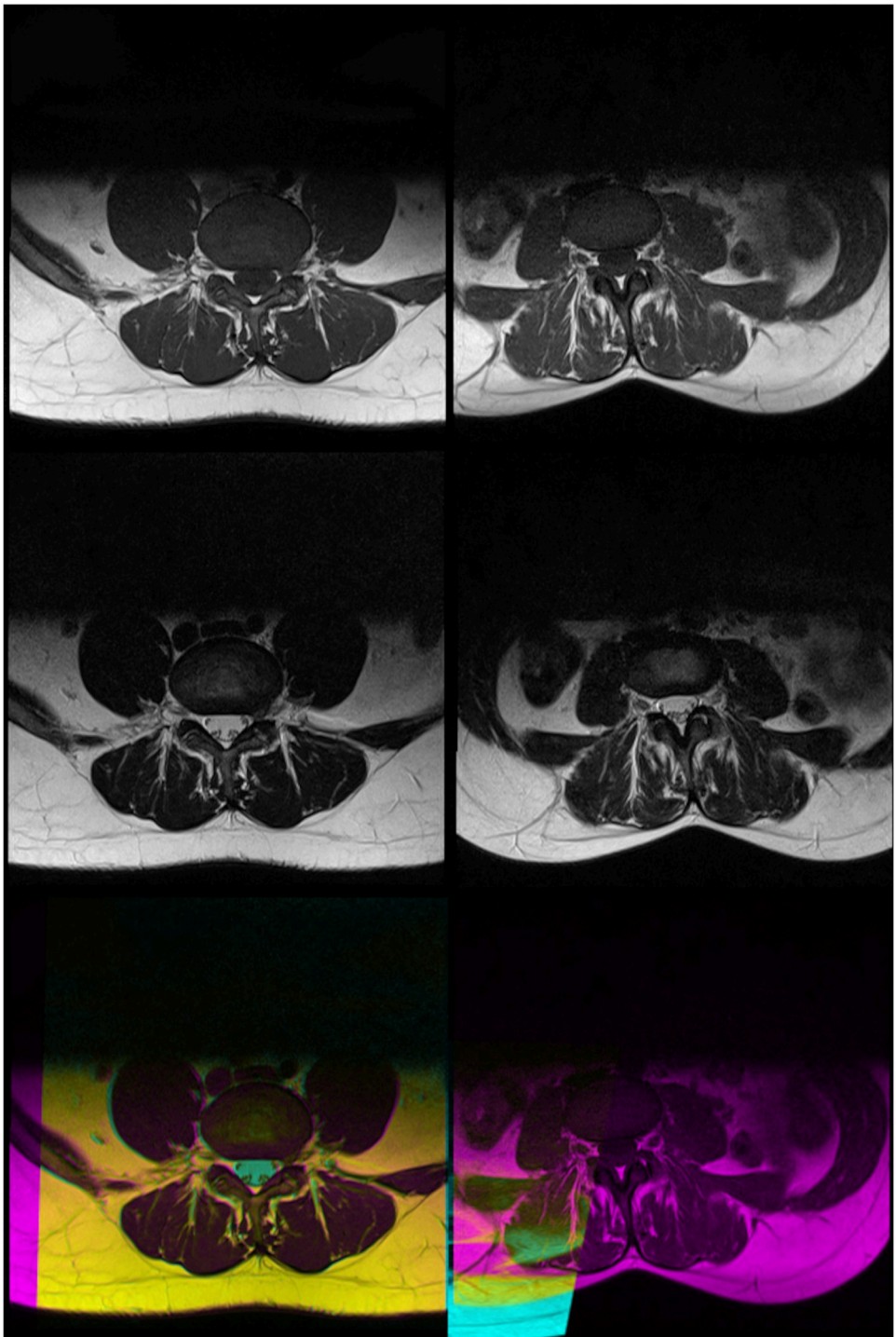

**Fig 5. Two example cases where the image registration process succeeded (left column) and failed (right column).** The top row shows the T1-weighted images, the middle row shows T2-weighted images, and the bottom row shows the resulting composite images after image registration.

**Table 2. Dataset sizes (number of images).**

| Class | Original Size (T1 and T2 pairs) | After Registration (Composite Images) | After Augmentation (Composite Images) |
|---|---|---|---|
| *best_d3* | 515 | 513 | 1,026 |
| *best_d3* | 515 | 513 | 1,026 |
| *best_d3* | 515 | 513 | 1,026 |
| *other_slices* | 7,391 | 7,371 | 1,539 |
| Total | 8,936 | 8,910 | 4,617 |

components whereas FA reduces the number of variables in the data by leveraging interdependencies between the observed variables. Feature selection techniques, on the other hand, work by ranking the untransformed features according to their importance. There are several feature ranking techniques in the literature, and in this study, we investigate three of the most popular ones including the Neighborhood Component Analysis (NCA) [40], the Minimum Redundancy Maximum Relevance (MRMR) [41], and the Chi-Square tests (CHI2) [42].

We use eleven DCNN architectures with each model pre-trained using the ImageNet database [31]. The list of the DCNNs and the summary of their architecture are shown in Table 3.

The image features produced by each DCNN are then reduced in length before being used to train several ML models and FC neural network models. In this study, we use the popular algorithm called FastICA [38] to realize the dimensionality reduction of the features using

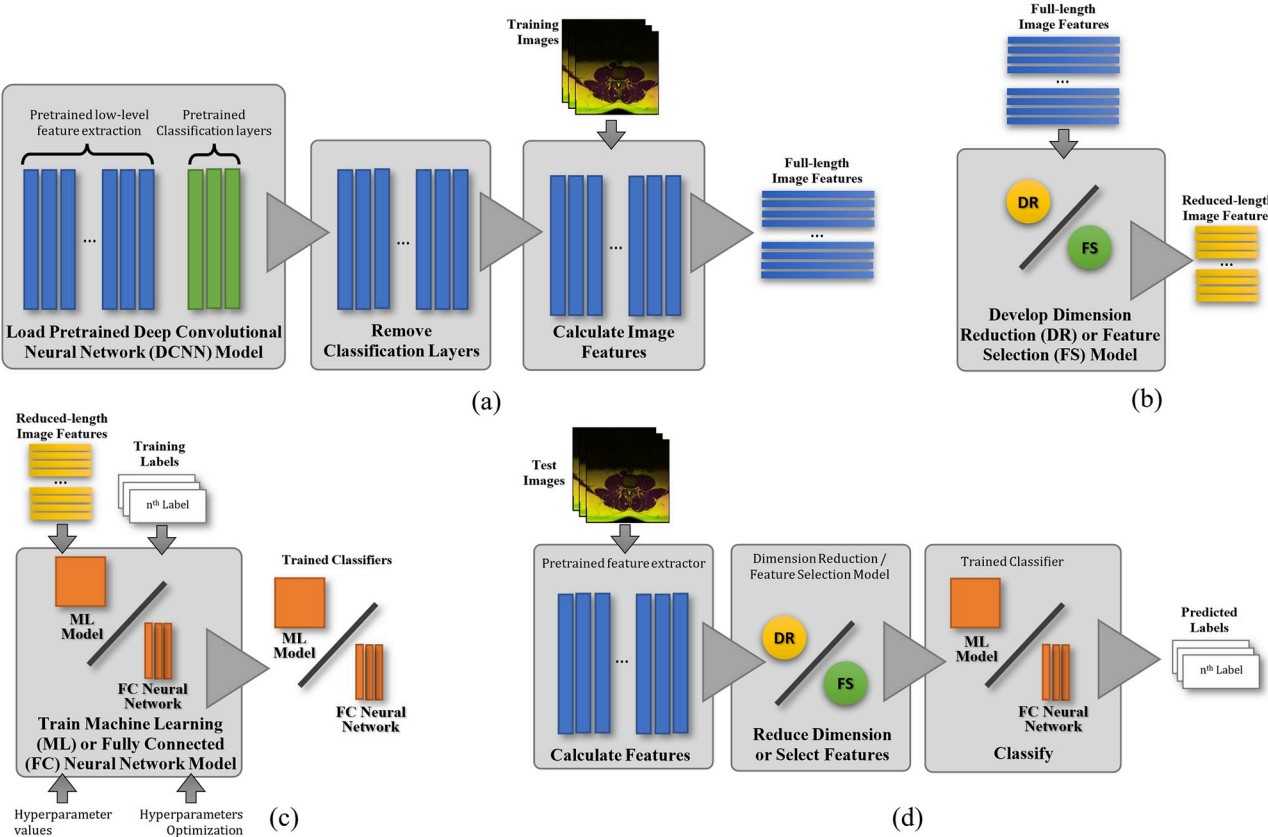

**Fig 6. A flowchart describing the methodology used where a) depicts the feature extraction step, b) depicts the DR and FS modeling step, c) depicts the ML and FC training step and d) depicts the inference/classification step.**

**Table 3. The list of the DCNNs used and the summary of their architecture.**

| Architecture | Depth (layers) | Parameters (millions) | Image Size (pixels) |
|---|---|---|---|
| **AlexNet (2012)** [43] | 8 | 61.0 | 227×227 |
| **DenseNet201 (2017)** [44] | 201 | 11.7 | 224×224 |
| **Inception-ResNet-V2 (2016)** [45] | 164 | 55.9 | 299×299 |
| **InceptionV3 (2016)** [46] | 48 | 23.9 | 299×299 |
| **MobileNetV2 (2018)** [47] | 53 | 3.5 | 224×224 |
| **ResNet18, ResNet50 and ResNet101 (2016)** [48] | 18, 50 and 101 | 11.7, 25.6 and 44.6 | 224×224 |
| **VGG16 and VGG19 (2015)** [49] | 16 and 19 | 138 and 144 | 224×224 |
| **Xception (2017)** [50] | 71 | 22.9 | 299×299 |

independent component analysis. Our approach also examines using the full-length (FL) image features, i.e., without applying any DR or FS technique, for the same purpose. In total, we have six sets of method combinations for each DCNN to compare with, namely DR-ML, DR-FC, FS-ML, FS-FC, FL-ML, and FL-FC. The DR-XX set consists of PCA-XX, FA-XX, and FastICA-XX whereas the FS-XX set consists of NCA-XX, MRMR-XX, and CHI2-XX, where XX denotes the classifier type which is either ML or FC. A tree diagram depicting the method combination is shown in Fig 7.

For the classification step, we use five different ML algorithms trained with hyperparameter optimization using a wide range of hyperparameter options and values. The ML algorithms that we used are K-Nearest Neighbor (KNN), Binary Decision Tree [51], Support Vector Machine (SVM) [52], Discriminant Analysis [53], and Ensemble of Tree Classifiers [52]. Each

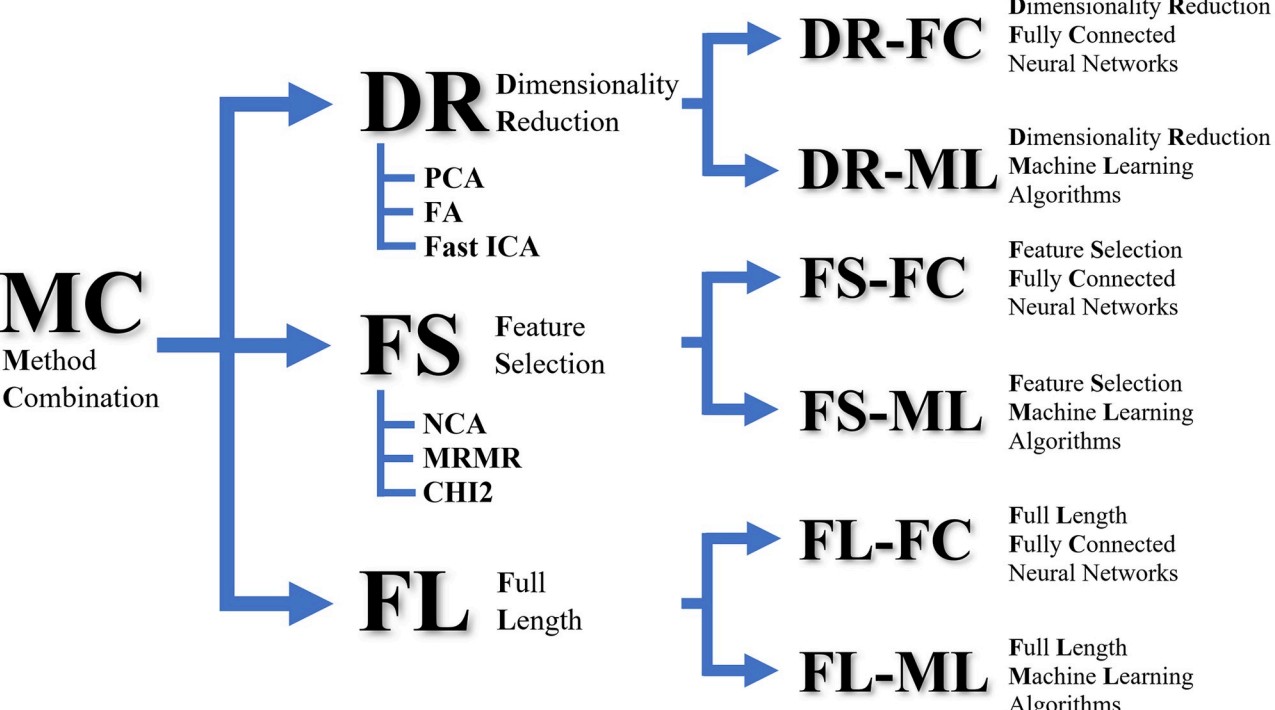

**Fig 7. A tree diagram depicting the method combination used in this study.**

**Table 4. The description of the hyperparameter optimization options and range of values for each ML learner.**

| Learner ID Name | Fixed Parameters | Optimized Hyperparameters |
|---|---|---|
| Fine KNN (FKNN) | Type: K-Nearest Neighbor<br>• Distance Metric: Euclidean<br>• Distance Weight: Equal<br>• Max node size: 50<br>• Prior probability: Empirical | • Number of Neighbors (1–10)<br>• Multiclass Decomposition Strategy (One-vs-One or One-vs-All)<br>• Input standardization (Yes or No) |
| Medium KNN (MKNN) | Type: K-Nearest Neighbor<br>• Distance Metric: Euclidean<br>• Distance Weight: Equal<br>• Max node size: 50<br>• Prior probability: Empirical | • Number of Neighbors (10–100)<br>• Multiclass Decomposition Strategy (One-vs-One or One-vs-All)<br>• Input standardization (Yes or No) |
| Coarse KNN (CKNN) | Type: K-Nearest Neighbor<br>• Distance Metric: Euclidean<br>• Distance Weight: Equal<br>• Max node size: 50<br>• Prior probability: Empirical | • Number of Neighbors (100–1000)<br>• Multiclass Decomposition Strategy [54] (One-vs-One or One-vs-All)<br>• Input standardization (Yes or No) |
| Cosine KNN (COSKNN) | Type: K-Nearest Neighbor<br>• Distance Metric: Cosine<br>• Distance Weight: Equal<br>• Max node size: 50<br>• Prior probability: Empirical | • Number of Neighbors (10–100)<br>• Multiclass Decomposition Strategy (One-vs-One or One-vs-All)<br>• Input standardization (Yes or No) |
| Cubic KNN (CUBKNN) | Type: K-Nearest Neighbor<br>• Distance Metric: Minkowski<br>• Distance Weight: Equal<br>• Max node size: 50<br>• Prior probability: Empirical<br>• Exponent: 3 | • Score transform (Yes or No)<br>• Number of Neighbors (10–100)<br>• Multiclass Decomposition Strategy (One-vs-One or One-vs-All)<br>• Input standardization (Yes or No) |
| Weighted KNN (WKNN) | Type: K-Nearest Neighbor<br>• Distance Metric: Euclidean<br>• Distance Weight: Squared Inverse $(1/d^2)$<br>• Max node size: 50<br>• Prior probability: Empirical | • Score transform (Yes or No)<br>• Number of Neighbors (10–100)<br>• Multiclass Decomposition Strategy (One-vs-One or One-vs-All)<br>• Input standardization (Yes or No) |
| Coarse Tree (CTREE) | Type: Binary Decision Tree [51]<br>• Min Leaf Size (1)<br>• Min Parent Size (10)<br>• Prior probability (Empirical)<br>• Score transform (None)<br>• Split Criterion (Gini-Simpson diversity index [55]) | • Max Number of Decision Splits (4–20)<br>• Multiclass Decomposition Strategy (One-vs-One or One-vs-All) |
| Medium Tree (MTREE) | Type: Binary Decision Tree<br>• Min Leaf Size (1)<br>• Min Parent Size (10)<br>• Prior probability (Empirical)<br>• Score transform (None)<br>Split Criterion (Gini-Simpson diversity index) | • Max Number of Decision Splits (20–100)<br>• Multiclass Decomposition Strategy (One-vs-One or One-vs-All) |
| Fine Tree (FTREE) | Type: Binary Decision Tree<br>• Min Leaf Size (1)<br>• Min Parent Size (10)<br>• Prior probability (Empirical)<br>• Score transform (None)<br>• Split Criterion (Gini-Simpson diversity index) | • Max Number of Decision Splits (100–1000)<br>• Multiclass Decomposition Strategy (One-vs-One or One-vs-All) |

*(Continued)*

**Table 4.** (Continued)

| Learner ID Name | Fixed Parameters | Optimized Hyperparameters |
|---|---|---|
| Linear SVM (LSVM) | Type: Support Vector Machine (SVM)<br>• Kernel Function: Linear | • Misclassification Penalty Cost (0.001–1000)<br>• Multiclass Decomposition Strategy (One-vs-One or One-vs-All)<br>• Input standardization (Yes or No) |
| Quadratic SVM (QSVM) | Type: Support Vector Machine (SVM)<br>• Kernel Function: Polynomial<br>• Polynomial Order: 2 | • Misclassification Penalty Cost (0.001–1000)<br>• Multiclass Decomposition Strategy (One-vs-One or One-vs-All)<br>• Input standardization (Yes or No) |
| Cubic SVM (CSVM) | Type: Support Vector Machine (SVM)<br>• Kernel Function: Polynomial<br>• Polynomial Order: 3 | • Misclassification Penalty Cost (0.001–1000)<br>• Multiclass Decomposition Strategy (One-vs-One or One-vs-All)<br>• Input standardization (Yes or No) |
| Fine Gaussian SVM (FGSVM) | Type: Support Vector Machine (SVM)<br>• Kernel Function: Gaussian | • Kernel Scale: (10–40)<br>• Misclassification Penalty Cost (0.001–1000)<br>• Multiclass Decomposition Strategy (One-vs-One or One-vs-All)<br>• Input standardization (Yes or No) |
| Medium Gaussian SVM (MGSVM) | Type: Support Vector Machine (SVM)<br>• Kernel Function: Gaussian | • Kernel Scale: (40–80)<br>• Misclassification Penalty Cost (0.001–1000)<br>• Multiclass Decomposition Strategy (One-vs-One or One-vs-All)<br>• Input standardization (Yes or No) |
| Coarse Gaussian SVM (CGSVM) | Type: Support Vector Machine (SVM)<br>• Kernel Function: Gaussian | • Kernel Scale: (80–160)<br>• Misclassification Penalty Cost (0.001–1000)<br>• Multiclass Decomposition Strategy (One-vs-One or One-vs-All)<br>• Input standardization (Yes or No) |
| Linear Discriminant (LD) | Type: Discriminant Analysis<br>• Discriminant Type: Pseudolinear | • Linear coefficient threshold (0.000001–1000)<br>• Regularization amount: (0–1)<br>• Multiclass Decomposition Strategy (One-vs-One or One-vs-All) |
| Quadratic Discriminant (QD) | Type: Discriminant Analysis<br>• Discriminant Type: Pseudo quadratic | • Regularization amount: (0–1)<br>• Multiclass Decomposition Strategy (One-vs-One or One-vs-All) |
| Bagged Trees Classifier (BAG) | Type: Ensemble of Tree Classifiers<br>• Method: Bootstrap Aggregation [56]<br>• Min Leaf Size (1)<br>• Min Parent Size (10)<br>• Prior probability (empirical)<br>• Score transform (none)<br>• Split Criterion (Gini-Simpson diversity index) | • Number of Learning Cycles: (10–50)<br>• Max Number of Splits: (3500–4500) |
| Boosted Trees Classifier (BOOST) | Type: Ensemble of Tree Classifiers<br>• Method: AdaBoostM2 [57]<br>• Min Leaf Size (1)<br>• Min Parent Size (10)<br>• Prior probability (empirical)<br>• Score transform (none)<br>• Split Criterion (Gini-Simpson diversity index) | • Learning rate (0.001–0.1)<br>• Number of Learning Cycles: (10–50)<br>• Max Number of Splits: (3500–4500) |
| Random Under Sampling (RUS) Boosting | Type: Ensemble of Tree Classifiers<br>• Random Under Sampling Boosting [58]<br>• Min Leaf Size (1)<br>• Min Parent Size (10)<br>• Prior probability (empirical)<br>• Score transform (none)<br>• Split Criterion (Gini-Simpson diversity index) | • Learning rate (0.001–0.1)<br>• Number of Learning Cycles: (10–50)<br>• Max Number of Splits: (3500–4500) |

**Table 5. The description of the hyperparameter optimization options and values for each FC learning optimization algorithm.**

| Classifier ID Name | Fixed Parameters | Optimized Hyperparameters |
|---|---|---|
| Stochastic Gradient Descent with Momentum (SGDM) | Optimizer Type: Stochastic Gradient Descent with Momentum<br>• Shuffle Every Epoch (True)<br>• Max Epochs (30)<br>• Constant learning rate per each training (as set by the Hyperparameters Optimization process)<br>• Validation frequency (3 iterations)<br>• All mini-batch lengths to match the longest mini-batch by zero-padding. | • Mini Batch Size (10–20)<br>• Initial Learning Rate (0.001–0.1)<br>• L2 Regularization Coefficient $(10^{-10}–10^{-2})$<br>• Momentum (0.8–0.98) |
| Root Mean Square Propagation (RMSP) | Optimizer: Root Mean Square Propagation<br>• Shuffle Every Epoch (True)<br>• Max Epochs (30)<br>• Constant learning rate per each training (as set by the Hyperparameters Optimization process)<br>• Validation frequency (3 iterations)<br>• All mini-batch lengths to match the longest mini-batch by zero-padding | • Mini Batch Size (10–20)<br>• Initial Learning Rate (0.001–0.1)<br>• L2 Regularization Coefficient $(10^{-10}–10^{-2})$<br>• Squared Gradient Decay Factor (0.8–0.98) |
| Adam Optimizer (ADAM) | Optimizer: Adam Optimizer<br>• Shuffle Every Epoch (True)<br>• Max Epochs (30)<br>• Constant learning rate per each training (as set by the Hyperparameters Optimization process)<br>• Validation frequency (3 iterations)<br>• All mini-batch lengths to match the longest mini-batch by zero-padding. | • Mini Batch Size (10–20)<br>• Initial Learning Rate (0.001–0.1)<br>• L2 Regularization Coefficient $(10^{-10}–10^{-2})$<br>• Gradient Decay Factor (0.8–0.98) |

of the algorithms may have more than one training setup. We group the different setups of hyperparameter optimization of all ML algorithms into 20 ML learners. The description of the hyperparameter optimization options and values for each ML learner is described in Table 4.

There are three training setups for the FC neural network, each corresponds to a different learning optimization algorithm, which is the Stochastic Gradient Descent with Momentum (SGDM) [59], the Root Mean Square Propagation (RMSP) [60], and the Adam (ADAM) [61] optimizers. The description of the hyperparameter optimization options and values for each ML learner is described in Table 5.

We used Bayesian optimization to search the hyperparameter space to find the best set of hyperparameter values for the classification model that produces the lowest output to the objective function which, in our case, is the misclassification rate. The Bayesian optimization method works by building a probability model of the objective function and using it to select the most promising hyperparameters to evaluate in the true objective function. The popular Expected Improvement (EI) method [62] was used as the acquisition function to guide how the hyperparameter space should be explored. The technique combines the predicted mean and the predicted variance generated by a Gaussian process model used during the optimization into a criterion that will direct the search. In general, any acquisition function needs to find a good trade-off between *exploitation* which focuses on searching the vicinity of the current best hyperparameter values, and *exploration* which pushes the search towards unexplored areas in the hyperparameter space. The EI method that we used was found to provide a good balance between exploitation and exploration [63]. Furthermore, an improvement technique as suggested in [64] was applied that modifies the behavior of the EI method when it is found to be overexploiting an area and allow it to escape a local objective function minimum.

With respect to the total number of methods in each combination shown in Fig 7, the DR-ML and FS-ML categories consist of 660 different methods. They are a combination of 11

**Table 6. The feature length of each DCNN model and DR/FS method combination.**

|  | PCA | FA | FastICA | NCA | MRMR | CHI2 | Full Length |
|---|---|---|---|---|---|---|---|
| **AlexNet** | 381 | 410 | 614 | 819 | 2048 | 1638 | 4096 |
| **DenseNet201** | 318 | 288 | 192 | 768 | 960 | 768 | 1920 |
| **Inception-ResNetv2** | 174 | 307 | 384 | 614 | 614 | 768 | 1536 |
| **Inceptionv3** | 377 | 205 | 205 | 819 | 819 | 819 | 2048 |
| **MobileNetv2** | 344 | 192 | 320 | 512 | 640 | 640 | 1280 |
| **ResNet18** | 158 | 102 | 128 | 256 | 256 | 256 | 512 |
| **ResNet50** | 305 | 307 | 205 | 1024 | 614 | 410 | 2048 |
| **ResNet101** | 274 | 410 | 205 | 1024 | 410 | 614 | 2048 |
| **VGG16** | 310 | 410 | 819 | 1638 | 2048 | 1638 | 4096 |
| **VGG19** | 311 | 410 | 410 | 1638 | 2048 | 1638 | 4096 |
| **Xception** | 338 | 410 | 410 | 614 | 819 | 1024 | 2048 |

The last column shows the full length of the DCNN features.

DCNNs, 3 DR and 3 FS techniques, and 20 ML learners. On the other hand, the DR-FC and FS-FC categories consist of 99 different methods. They are a combination of 11 DCNNs, 3 DR and 3 FS techniques, and 3 FC optimization algorithms. It is important to note that the traditional transfer learning approach as depicted in Fig 4 is the FL-FC method.

## 3. Experimental results, analysis, and discussion

### 3.1. Feature dimension reduction and feature selection

Before we could proceed with implementing the image classification processes, we need to determine the best feature length for each combination of DCNN model and DR/FS method. For PCA we use the popular 0.95 explained-variance threshold [65]. For the others, we tested different length percentages (ranging from 5 to 25% in 5% increments) for each classifier and identify one that gives the best overall accuracy. We select one representative length for each DCNN-DR/FS combination by calculating the average value over all classifiers. The resulting feature length values for each DCNN-DR/FS combination are shown in Table 6.

### 3.2. Performance metrics

The classification performance of each method is measured using four performance metrics namely the overall Accuracy, Precision, Recall, and F1-Score. The overall Accuracy, denoted as $A$, is the ratio of the number of correctly classified images and the total number of test images. Using the standard notations of true positive ($tp$), true negative ($tn$), false positive ($fp$), and false negative ($fn$), the metrics are calculated as:

$$A = \frac{tp + tn}{tp + tn + fp + fn}$$

Since the Accuracy metric could provide a misleading assessment of a method's performance on an imbalanced dataset, we also employ the other three metrics to provide us with a more complete picture of the method's performance. The Precision, Recall, and their

harmonic mean F1-Score metrics are calculated as:

$$P = \frac{1}{C}\sum_{i}^{C} P_i$$

$$R = \frac{1}{C}\sum_{i}^{C} R_i$$

$$F = \frac{2}{C}\sum_{i}^{C} \frac{P_i \cdot R_i}{P_i + R_i}$$

Where $P_i$, $R_i$, and $F_i$ denote the respective class-based metrics of the $i^{th}$ class, which are the class precision, the class recall, and the class F1-Score, respectively. The notation $i$, where $i \in \{1, 2, 3, 4\}$, is the respective index of each of the four classes. These class-based metrics provide a measure of each method's performance on each individual class and are calcu-

**Table 7. The best ML learner for each combination method and its average classification performance (using Accuracy metric).**

|  | PCA-ML | FA-ML | FastICA-ML | NCA-ML | MRMR-ML | CHI2-ML | FL-ML |
|---|---|---|---|---|---|---|---|
| **AlexNet** | 0.96 | 0.95 | 0.96 | 0.96 | 0.96 | 0.96 | 0.95 |
|  | FGSVM | CSVM | FGSVM | FGSVM | FGSVM | FGSVM | FGSVM |
| **DenseNet201** | 0.96 | 0.97 | 0.96 | 0.97 | 0.97 | 0.97 | 0.97 |
|  | CSVM | FGSVM | FGSVM | FGSVM | FGSVM | FGSVM | FGSVM |
| **Inception-ResNetv2** | 0.95 | 0.96 | 0.96 | 0.95 | 0.95 | 0.95 | 0.95 |
|  | FGSVM | FGSVM | FGSVM | FGSVM | FGSVM | FGSVM | FGSVM |
| **Inceptionv3** | 0.96 | 0.95 | 0.96 | 0.97 | 0.96 | 0.96 | 0.96 |
|  | FGSVM | FGSVM | FGSVM | FGSVM | FGSVM | FGSVM | FGSVM |
| **MobileNetv2** | 0.96 | 0.95 | 0.96 | 0.96 | 0.96 | 0.96 | 0.96 |
|  | FGSVM | CSVM | FGSVM | FGSVM | FGSVM | FGSVM | FGSVM |
| **ResNet18** | 0.94 | 0.93 | 0.94 | 0.94 | 0.94 | 0.94 | 0.95 |
|  | FGSVM | BAG | FGSVM | FGSVM | FGSVM | FGSVM | FGSVM |
| **ResNet50** | 0.96 | 0.95 | 0.96 | 0.97 | 0.95 | 0.96 | 0.96 |
|  | FGSVM | FGSVM | FGSVM | FGSVM | FGSVM | FGSVM | FGSVM |
| **ResNet101** | 0.96 | 0.96 | 0.96 | 0.95 | 0.95 | 0.96 | 0.96 |
|  | FGSVM | FGSVM | FGSVM | FGSVM | FGSVM | FGSVM | FGSVM |
| **VGG16** | 0.95 | 0.95 | 0.96 | 0.94 | 0.94 | 0.94 | 0.95 |
|  | FGSVM | FGSVM | FGSVM | FGSVM | FGSVM | FGSVM | FGSVM |
| **VGG19** | 0.96 | 0.96 | 0.96 | 0.95 | 0.95 | 0.95 | 0.94 |
|  | FGSVM | FGSVM | FGSVM | FGSVM | FGSVM | FGSVM | FGSVM |
| **Xception** | 0.96 | 0.96 | 0.96 | 0.95 | 0.96 | 0.96 | 0.96 |
|  | FGSVM | FGSVM | FGSVM | FGSVM | FGSVM | FGSVM | FGSVM |

**Table 8. The best ML learner for each combination method and its average classification performance (using Precision metric).**

|  | PCA-ML | FA-ML | FastICA-ML | NCA-ML | MRMR-ML | CHI2-ML | FL-ML |
|---|---|---|---|---|---|---|---|
| **AlexNet** | 0.96 | 0.95 | 0.96 | 0.96 | 0.95 | 0.95 | 0.95 |
|  | FGSVM | CSVM | FGSVM | FGSVM | FGSVM | FGSVM | FGSVM |
| **DenseNet201** | 0.95 | 0.97 | 0.96 | 0.97 | 0.97 | 0.97 | 0.97 |
|  | FGSVM | FGSVM | FGSVM | FGSVM | FGSVM | FGSVM | FGSVM |
| **Inception-ResNetv2** | 0.95 | 0.95 | 0.96 | 0.95 | 0.95 | 0.95 | 0.95 |
|  | FGSVM | FGSVM | FGSVM | FGSVM | FGSVM | FGSVM | FGSVM |
| **Inceptionv3** | 0.96 | 0.95 | 0.96 | 0.96 | 0.96 | 0.96 | 0.96 |
|  | FGSVM | FGSVM | FGSVM | FGSVM | FGSVM | FGSVM | FGSVM |
| **MobileNetv2** | 0.96 | 0.95 | 0.96 | 0.96 | 0.95 | 0.96 | 0.96 |
|  | FGSVM | CSVM | FGSVM | FGSVM | FGSVM | FGSVM | FGSVM |
| **ResNet18** | 0.94 | 0.93 | 0.94 | 0.94 | 0.94 | 0.94 | 0.95 |
|  | FGSVM | BAG | FGSVM | FGSVM | FGSVM | FGSVM | FGSVM |
| **ResNet50** | 0.95 | 0.95 | 0.96 | 0.97 | 0.95 | 0.96 | 0.96 |
|  | FGSVM | FGSVM | FGSVM | FGSVM | FGSVM | FGSVM | FGSVM |
| **ResNet101** | 0.96 | 0.96 | 0.95 | 0.95 | 0.95 | 0.95 | 0.95 |
|  | FGSVM | FGSVM | FGSVM | FGSVM | FGSVM | FGSVM | FGSVM |
| **VGG16** | 0.94 | 0.95 | 0.96 | 0.94 | 0.94 | 0.94 | 0.95 |
|  | FGSVM | FGSVM | FGSVM | FGSVM | FGSVM | FGSVM | FGSVM |
| **VGG19** | 0.96 | 0.96 | 0.96 | 0.94 | 0.95 | 0.95 | 0.94 |
|  | FGSVM | FGSVM | FGSVM | FGSVM | FGSVM | FGSVM | FGSVM |
| **Xception** | 0.96 | 0.96 | 0.96 | 0.95 | 0.96 | 0.96 | 0.96 |
|  | FGSVM | FGSVM | FGSVM | FGSVM | FGSVM | FGSVM | FGSVM |

**Table 9. The best ML learner for each combination method and its average classification performance (using Recall metric).**

|  | PCA-ML | FA-ML | FastICA-ML | NCA-ML | MRMR-ML | CHI2-ML | FL-ML |
|---|---|---|---|---|---|---|---|
| **AlexNet** | 0.96 | 0.96 | 0.96 | 0.96 | 0.96 | 0.96 | 0.96 |
|  | FGSVM | CSVM | FGSVM | FGSVM | FGSVM | FGSVM | FGSVM |
| **DenseNet201** | 0.96 | 0.97 | 0.97 | 0.97 | 0.97 | 0.97 | 0.98 |
|  | CSVM | FGSVM | FGSVM | FGSVM | FGSVM | FGSVM | FGSVM |
| **Inception-ResNetv2** | 0.96 | 0.96 | 0.96 | 0.96 | 0.96 | 0.96 | 0.96 |
|  | FGSVM | FGSVM | FGSVM | FGSVM | FGSVM | FGSVM | FGSVM |
| **Inceptionv3** | 0.96 | 0.96 | 0.96 | 0.97 | 0.96 | 0.97 | 0.97 |
|  | FGSVM | FGSVM | FGSVM | FGSVM | FGSVM | FGSVM | FGSVM |
| **MobileNetv2** | 0.96 | 0.96 | 0.96 | 0.96 | 0.96 | 0.97 | 0.97 |
|  | FGSVM | CSVM | FGSVM | FGSVM | FGSVM | FGSVM | FGSVM |
| **ResNet18** | 0.95 | 0.94 | 0.95 | 0.95 | 0.95 | 0.95 | 0.96 |
|  | CSVM | BAG | FGSVM | FGSVM | FGSVM | FGSVM | FGSVM |
| **ResNet50** | 0.96 | 0.96 | 0.96 | 0.97 | 0.96 | 0.97 | 0.97 |
|  | FGSVM | CSVM | FGSVM | FGSVM | CSVM | FGSVM | FGSVM |
| **ResNet101** | 0.96 | 0.96 | 0.96 | 0.96 | 0.96 | 0.96 | 0.96 |
|  | FGSVM | FGSVM | FGSVM | FGSVM | FGSVM | FGSVM | FGSVM |
| **VGG16** | 0.95 | 0.95 | 0.96 | 0.95 | 0.95 | 0.95 | 0.96 |
|  | FGSVM | FGSVM | FGSVM | FGSVM | FGSVM | FGSVM | FGSVM |
| **VGG19** | 0.97 | 0.96 | 0.96 | 0.95 | 0.96 | 0.95 | 0.95 |
|  | FGSVM | FGSVM | FGSVM | FGSVM | FGSVM | FGSVM | FGSVM |
| **Xception** | 0.96 | 0.96 | 0.96 | 0.96 | 0.96 | 0.96 | 0.96 |
|  | FGSVM | FGSVM | FGSVM | FGSVM | FGSVM | FGSVM | FGSVM |

**Table 10.  The best ML learner for each combination method and its average classification performance (using F1-Score metric).**

|  | PCA-ML | FA-ML | FastICA-ML | NCA-ML | MRMR-ML | CHI2-ML | FL-ML |
|---|---|---|---|---|---|---|---|
| **AlexNet** | 0.96 | 0.95 | 0.96 | 0.96 | 0.96 | 0.96 | 0.96 |
|  | FGSVM | CSVM | FGSVM | FGSVM | FGSVM | FGSVM | FGSVM |
| **DenseNet201** | 0.96 | 0.97 | 0.96 | 0.97 | 0.97 | 0.97 | 0.97 |
|  | CSVM | FGSVM | FGSVM | FGSVM | FGSVM | FGSVM | FGSVM |
| **Inception-ResNetv2** | 0.95 | 0.96 | 0.96 | 0.95 | 0.95 | 0.95 | 0.95 |
|  | FGSVM | FGSVM | FGSVM | FGSVM | FGSVM | FGSVM | FGSVM |
| **Inceptionv3** | 0.96 | 0.96 | 0.96 | 0.97 | 0.96 | 0.96 | 0.96 |
|  | FGSVM | FGSVM | FGSVM | FGSVM | FGSVM | FGSVM | FGSVM |
| **MobileNetv2** | 0.96 | 0.95 | 0.96 | 0.96 | 0.96 | 0.96 | 0.96 |
|  | FGSVM | CSVM | FGSVM | FGSVM | FGSVM | FGSVM | FGSVM |
| **ResNet18** | 0.94 | 0.94 | 0.94 | 0.94 | 0.94 | 0.95 | 0.96 |
|  | FGSVM | BAG | FGSVM | FGSVM | FGSVM | FGSVM | FGSVM |
| **ResNet50** | 0.96 | 0.95 | 0.96 | 0.97 | 0.95 | 0.96 | 0.96 |
|  | FGSVM | FGSVM | FGSVM | FGSVM | FGSVM | FGSVM | FGSVM |
| **ResNet101** | 0.96 | 0.96 | 0.96 | 0.95 | 0.95 | 0.96 | 0.96 |
|  | FGSVM | FGSVM | FGSVM | FGSVM | FGSVM | FGSVM | FGSVM |
| **VGG16** | 0.95 | 0.95 | 0.96 | 0.95 | 0.94 | 0.94 | 0.95 |
|  | FGSVM | FGSVM | FGSVM | FGSVM | FGSVM | FGSVM | FGSVM |
| **VGG19** | 0.96 | 0.96 | 0.96 | 0.95 | 0.95 | 0.95 | 0.94 |
|  | FGSVM | FGSVM | FGSVM | FGSVM | FGSVM | FGSVM | FGSVM |
| **Xception** | 0.96 | 0.96 | 0.96 | 0.95 | 0.96 | 0.96 | 0.96 |
|  | FGSVM | FGSVM | FGSVM | FGSVM | FGSVM | FGSVM | FGSVM |

lated as:

$$P_i = \frac{tp_i}{tp_i + fp_i}$$

$$R_i = \frac{tp_i}{tp_i + fn_i}$$

$$F_i = 2 \times \frac{P_i \cdot R_i}{P_i + R_i}$$

## 3.3. Experimental results, analysis, and discussion

We implemented each method combination 20 times, each with a different combination of training and test sets, to provide us with statistically representative results. The average of each performance metric over the 20 repeats of each method combination is calculated. The ML learner or FC optimizer that produces the highest performance of each method combination is then identified. The results are shown in Tables 7–14.

Through observation of the experimental results shown in those tables, we can deduce several findings. Firstly, the performance of DR-ML methods is very good and stable with a range of values from 0.93 to 0.97. The performance of FS-ML methods is also very good and stable with a range of values from 0.94 to 0.97. Likewise, the performance of FL-ML methods is also very good and stable with a range of values from 0.94 to 0.98. On the other hand, the performance of DR-FC, FS-FC, and FL-FC method combinations is generally poorer and more unpredictable. This can be seen from the bigger range in performance the method

**Table 11. The best FC learning optimizer for each combination method and its average classification performance (using Accuracy metric).**

| | PCA-FC | FA-FC | FastICA-FC | NCA-FC | MRMR-FC | CHI2-FC | FL-FC |
|---|---|---|---|---|---|---|---|
| **AlexNet** | 0.85 | 0.82 | 0.89 | 0.88 | 0.96 | 0.92 | 0.96 |
| | RMSP | SGDM | ADAM | RMSP | ADAM | ADAM | ADAM |
| **DenseNet201** | 0.88 | 0.88 | 0.77 | 0.87 | 0.84 | 0.82 | 0.93 |
| | RMSP | RMSP | ADAM | RMSP | RMSP | RMSP | SGDM |
| **Inception-ResNetv2** | 0.73 | 0.82 | 0.78 | 0.78 | 0.78 | 0.72 | 0.88 |
| | ADAM | SGDM | ADAM | RMSP | RMSP | RMSP | RMSP |
| **Inceptionv3** | 0.84 | 0.75 | 0.68 | 0.86 | 0.85 | 0.84 | 0.90 |
| | RMSP | SGDM | RMSP | RMSP | RMSP | RMSP | RMSP |
| **MobileNetv2** | 0.89 | 0.82 | 0.81 | 0.88 | 0.85 | 0.88 | 0.95 |
| | RMSP | SGDM | ADAM | RMSP | RMSP | SGDM | RMSP |
| **ResNet18** | 0.78 | 0.74 | 0.67 | 0.76 | 0.78 | 0.75 | 0.78 |
| | SGDM | ADAM | ADAM | ADAM | RMSP | RMSP | ADAM |
| **ResNet50** | 0.87 | 0.83 | 0.75 | 0.90 | 0.82 | 0.83 | 0.96 |
| | ADAM | SGDM | ADAM | RMSP | RMSP | RMSP | ADAM |
| **ResNet101** | 0.83 | 0.91 | 0.72 | 0.86 | 0.78 | 0.85 | 0.93 |
| | ADAM | SGDM | ADAM | ADAM | RMSP | RMSP | ADAM |
| **VGG16** | 0.82 | 0.87 | 0.85 | 0.96 | 0.96 | 0.94 | 0.94 |
| | ADAM | RMSP | ADAM | ADAM | ADAM | ADAM | ADAM |
| **VGG19** | 0.86 | 0.90 | 0.80 | 0.97 | 0.95 | 0.91 | 0.97 |
| | RMSP | RMSP | ADAM | ADAM | ADAM | ADAM | ADAM |
| **Xception** | 0.84 | 0.88 | 0.77 | 0.77 | 0.84 | 0.85 | 0.87 |
| | ADAM | SGDM | ADAM | ADAM | RMSP | RMSP | RMSP |

**Table 12. The best FC learning optimizer for each combination method and its average classification performance (using Precision metric).**

| | PCA-FC | FA-FC | FastICA-FC | NCA-FC | MRMR-FC | CHI2-FC | FL-FC |
|---|---|---|---|---|---|---|---|
| **AlexNet** | 0.84 | 0.81 | 0.89 | 0.89 | 0.96 | 0.93 | 0.96 |
| | RMSP | ADAM | ADAM | RMSP | ADAM | ADAM | ADAM |
| **DenseNet201** | 0.88 | 0.87 | 0.80 | 0.88 | 0.86 | 0.82 | 0.93 |
| | RMSP | RMSP | ADAM | RMSP | ADAM | RMSP | SGDM |
| **Inception-ResNetv2** | 0.72 | 0.81 | 0.81 | 0.81 | 0.77 | 0.81 | 0.87 |
| | ADAM | SGDM | ADAM | ADAM | RMSP | RMSP | RMSP |
| **Inceptionv3** | 0.83 | 0.74 | 0.75 | 0.88 | 0.86 | 0.85 | 0.92 |
| | RMSP | SGDM | RMSP | RMSP | RMSP | RMSP | RMSP |
| **MobileNetv2** | 0.89 | 0.81 | 0.84 | 0.89 | 0.87 | 0.87 | 0.95 |
| | RMSP | SGDM | ADAM | RMSP | ADAM | SGDM | RMSP |
| **ResNet18** | 0.77 | 0.72 | 0.71 | 0.76 | 0.77 | 0.75 | 0.82 |
| | SGDM | ADAM | RMSP | ADAM | RMSP | RMSP | ADAM |
| **ResNet50** | 0.86 | 0.83 | 0.78 | 0.91 | 0.87 | 0.82 | 0.96 |
| | ADAM | RMSP | ADAM | RMSP | ADAM | RMSP | ADAM |
| **ResNet101** | 0.82 | 0.90 | 0.74 | 0.86 | 0.80 | 0.85 | 0.95 |
| | ADAM | SGDM | ADAM | ADAM | RMSP | RMSP | RMSP |
| **VGG16** | 0.82 | 0.86 | 0.87 | 0.95 | 0.96 | 0.95 | 0.96 |
| | ADAM | RMSP | ADAM | ADAM | ADAM | ADAM | ADAM |
| **VGG19** | 0.85 | 0.89 | 0.83 | 0.97 | 0.94 | 0.93 | 0.98 |
| | RMSP | RMSP | ADAM | ADAM | ADAM | ADAM | ADAM |
| **Xception** | 0.84 | 0.88 | 0.80 | 0.77 | 0.83 | 0.85 | 0.87 |
| | ADAM | SGDM | ADAM | ADAM | RMSP | ADAM | RMSP |

**Table 13. The best FC learning optimizer for each combination method and its average classification performance (using Recall metric).**

|  | PCA-FC | FA-FC | FastICA-FC | NCA-FC | MRMR-FC | CHI2-FC | FL-FC |
|---|---|---|---|---|---|---|---|
| **AlexNet** | 0.85 | 0.83 | 0.87 | 0.89 | 0.97 | 0.90 | 0.96 |
|  | RMSP | SGDM | ADAM | ADAM | ADAM | ADAM | ADAM |
| **DenseNet201** | 0.89 | 0.88 | 0.74 | 0.86 | 0.84 | 0.80 | 0.93 |
|  | RMSP | RMSP | ADAM | RMSP | SGDM | RMSP | SGDM |
| **Inception-ResNetv2** | 0.73 | 0.82 | 0.75 | 0.76 | 0.76 | 0.72 | 0.89 |
|  | ADAM | SGDM | ADAM | RMSP | RMSP | ADAM | RMSP |
| **Inceptionv3** | 0.84 | 0.75 | 0.64 | 0.85 | 0.84 | 0.84 | 0.89 |
|  | RMSP | SGDM | ADAM | RMSP | RMSP | ADAM | ADAM |
| **MobileNetv2** | 0.88 | 0.81 | 0.79 | 0.87 | 0.87 | 0.90 | 0.95 |
|  | RMSP | SGDM | ADAM | RMSP | RMSP | RMSP | RMSP |
| **ResNet18** | 0.77 | 0.72 | 0.62 | 0.73 | 0.76 | 0.73 | 0.76 |
|  | SGDM | ADAM | ADAM | ADAM | RMSP | RMSP | SGDM |
| **ResNet50** | 0.87 | 0.83 | 0.71 | 0.89 | 0.82 | 0.82 | 0.97 |
|  | ADAM | RMSP | ADAM | RMSP | RMSP | RMSP | ADAM |
| **ResNet101** | 0.83 | 0.92 | 0.68 | 0.88 | 0.79 | 0.85 | 0.94 |
|  | ADAM | SGDM | ADAM | ADAM | ADAM | RMSP | ADAM |
| **VGG16** | 0.83 | 0.87 | 0.82 | 0.96 | 0.96 | 0.93 | 0.92 |
|  | ADAM | RMSP | ADAM | ADAM | ADAM | ADAM | ADAM |
| **VGG19** | 0.86 | 0.89 | 0.78 | 0.97 | 0.96 | 0.90 | 0.96 |
|  | RMSP | RMSP | ADAM | ADAM | ADAM | ADAM | ADAM |
| **Xception** | 0.85 | 0.89 | 0.74 | 0.76 | 0.84 | 0.85 | 0.86 |
|  | ADAM | ADAM | ADAM | ADAM | RMSP | RMSP | ADAM |

**Table 14. The best FC learning optimizer for each combination method and its average classification performance (using F1-Score metric).**

|  | PCA-FC | FA-FC | FastICA-FC | NCA-FC | MRMR-FC | CHI2-FC | FL-FC |
|---|---|---|---|---|---|---|---|
| **AlexNet** | 0.84 | 0.81 | 0.88 | 0.88 | 0.96 | 0.91 | 0.96 |
|  | RMSP | RMSP | ADAM | ADAM | ADAM | ADAM | ADAM |
| **DenseNet201** | 0.88 | 0.87 | 0.75 | 0.86 | 0.83 | 0.81 | 0.93 |
|  | RMSP | RMSP | ADAM | RMSP | SGDM | RMSP | SGDM |
| **Inception-ResNetv2** | 0.72 | 0.81 | 0.77 | 0.76 | 0.76 | 0.70 | 0.88 |
|  | ADAM | SGDM | ADAM | RMSP | RMSP | ADAM | RMSP |
| **Inceptionv3** | 0.84 | 0.74 | 0.66 | 0.86 | 0.85 | 0.83 | 0.89 |
|  | RMSP | SGDM | RMSP | RMSP | RMSP | RMSP | RMSP |
| **MobileNetv2** | 0.88 | 0.81 | 0.80 | 0.87 | 0.84 | 0.87 | 0.95 |
|  | RMSP | SGDM | ADAM | RMSP | RMSP | RMSP | RMSP |
| **ResNet18** | 0.77 | 0.72 | 0.63 | 0.73 | 0.76 | 0.73 | 0.76 |
|  | SGDM | ADAM | ADAM | ADAM | RMSP | RMSP | SGDM |
| **ResNet50** | 0.86 | 0.82 | 0.73 | 0.89 | 0.82 | 0.81 | 0.96 |
|  | ADAM | RMSP | ADAM | RMSP | RMSP | RMSP | ADAM |
| **ResNet101** | 0.82 | 0.90 | 0.69 | 0.86 | 0.77 | 0.85 | 0.93 |
|  | ADAM | SGDM | ADAM | ADAM | ADAM | RMSP | RMSP |
| **VGG16** | 0.82 | 0.86 | 0.84 | 0.96 | 0.96 | 0.94 | 0.94 |
|  | ADAM | RMSP | ADAM | ADAM | ADAM | ADAM | ADAM |
| **VGG19** | 0.85 | 0.89 | 0.80 | 0.97 | 0.95 | 0.91 | 0.97 |
|  | RMSP | RMSP | ADAM | ADAM | ADAM | ADAM | ADAM |
| **Xception** | 0.84 | 0.88 | 0.76 | 0.76 | 0.83 | 0.84 | 0.86 |
|  | ADAM | SGDM | ADAM | ADAM | RMSP | RMSP | RMSP |

**Table 15. Summary of the classification performance of ML learners using features from 11 DCNNs.**

|  | DR-ML | | | | FS-ML | | | | FL-ML | | | |
|---|---|---|---|---|---|---|---|---|---|---|---|---|
|  | *A* | *P* | *R* | *F* | *A* | *P* | *R* | *F* | *A* | *P* | *R* | *F* |
| Min | 0.93 | 0.93 | 0.94 | 0.94 | 0.94 | 0.94 | 0.95 | 0.94 | 0.94 | 0.94 | 0.95 | 0.94 |
| Max | 0.97 | 0.97 | 0.97 | 0.97 | 0.97 | 0.97 | 0.97 | 0.97 | 0.97 | 0.97 | 0.98 | 0.97 |
| Mean | 0.96 | 0.95 | 0.96 | 0.96 | 0.95 | 0.95 | 0.96 | 0.96 | 0.96 | 0.96 | 0.96 | 0.96 |

The table header A, P. R, and F are the shorthand of Accuracy, Precision, Recall, and F1-Score, respectively.

**Table 16. Summary of the classification performance of FC neural networks using features from 11 DCNNs.**

|  | DR-FC | | | | FS-FC | | | | FL-FC | | | |
|---|---|---|---|---|---|---|---|---|---|---|---|---|
|  | *A* | *P* | *R* | *F* | *A* | *P* | *R* | *F* | *A* | *P* | *R* | *F* |
| Min | 0.67 | 0.71 | 0.62 | 0.63 | 0.72 | 0.75 | 0.72 | 0.70 | 0.78 | 0.82 | 0.76 | 0.76 |
| Max | 0.91 | 0.90 | 0.92 | 0.90 | 0.97 | 0.97 | 0.97 | 0.97 | 0.97 | 0.98 | 0.97 | 0.97 |
| Mean | 0.81 | 0.82 | 0.80 | 0.80 | 0.85 | 0.86 | 0.85 | 0.85 | 0.91 | 0.92 | 0.91 | 0.91 |

The table header A, P. R, and F are the shorthand of Accuracy, Precision, Recall, and F1-Score, respectively.

**Table 17. Summary of classification performance of ML methods for each DCNN.**

|  | Accuracy | | | Precision | | | Recall | | | F1-Score | | |
|---|---|---|---|---|---|---|---|---|---|---|---|---|
|  | Min | Max | Mean | Min | Max | Mean | Min | Max | Mean | Min | Max | Mean |
| AlexNet | 0.95 | 0.96 | 0.96 | **0.95** | 0.96 | 0.95 | **0.96** | 0.96 | 0.96 | 0.95 | 0.96 | 0.96 |
| DenseNet201 | **0.96** | **0.97** | **0.97** | 0.95 | **0.97** | **0.96** | 0.96 | **0.98** | **0.97** | **0.96** | **0.97** | **0.97** |
| Inception-ResNetv2 | 0.95 | 0.96 | 0.95 | **0.95** | 0.96 | 0.95 | **0.96** | 0.96 | 0.96 | 0.95 | 0.96 | 0.95 |
| Inceptionv3 | 0.95 | 0.96 | 0.96 | **0.95** | 0.96 | **0.96** | **0.96** | 0.97 | 0.96 | 0.95 | **0.97** | 0.96 |
| MobileNetv2 | 0.95 | 0.96 | 0.96 | **0.95** | 0.96 | **0.96** | **0.96** | 0.97 | 0.96 | 0.95 | 0.96 | 0.96 |
| ResNet18 | 0.93 | 0.95 | 0.94 | 0.93 | 0.95 | 0.94 | 0.94 | 0.96 | 0.95 | 0.94 | 0.95 | 0.94 |
| ResNet50 | 0.95 | **0.97** | 0.96 | **0.95** | **0.97** | **0.96** | **0.96** | 0.97 | 0.96 | 0.95 | **0.97** | 0.96 |
| ResNet101 | 0.95 | 0.96 | 0.95 | **0.95** | 0.96 | 0.95 | **0.96** | 0.96 | 0.96 | 0.95 | 0.96 | 0.96 |
| VGG16 | 0.94 | 0.96 | 0.95 | 0.94 | 0.96 | 0.95 | 0.95 | 0.96 | 0.95 | 0.94 | 0.96 | 0.95 |
| VGG19 | 0.94 | 0.96 | 0.95 | 0.94 | 0.96 | 0.95 | 0.95 | 0.96 | 0.96 | 0.94 | 0.96 | 0.95 |
| Xception | 0.95 | 0.96 | 0.96 | **0.95** | 0.96 | **0.96** | **0.96** | 0.96 | 0.96 | 0.95 | 0.96 | 0.96 |

The maximum value in each column is marked in bold text.

combinations produce. Therefore, we can say that ML approaches are better and more predictable than FC approaches. Secondly, we find that Fine Gaussian SVM (FGSVM) is the best learner to use in the ML category as it consistently produces the highest performance in all four metrics. This learner uses the Support Vector Machine algorithm with a short Gaussian kernel. On the other hand, there is no clear best optimizer in the FC category as both ADAM and RMSP optimizers are relatively equal. However, when full-length features from ResNet50, ResNet101, VGG16, and VGG19 DCNNs were used ADAM optimizer produces generally higher performance than the other two.

We also found that applying DR or FS to the features has little impact on the performance when ML classifiers were used. However, this is not the case when FC Neural Networks were

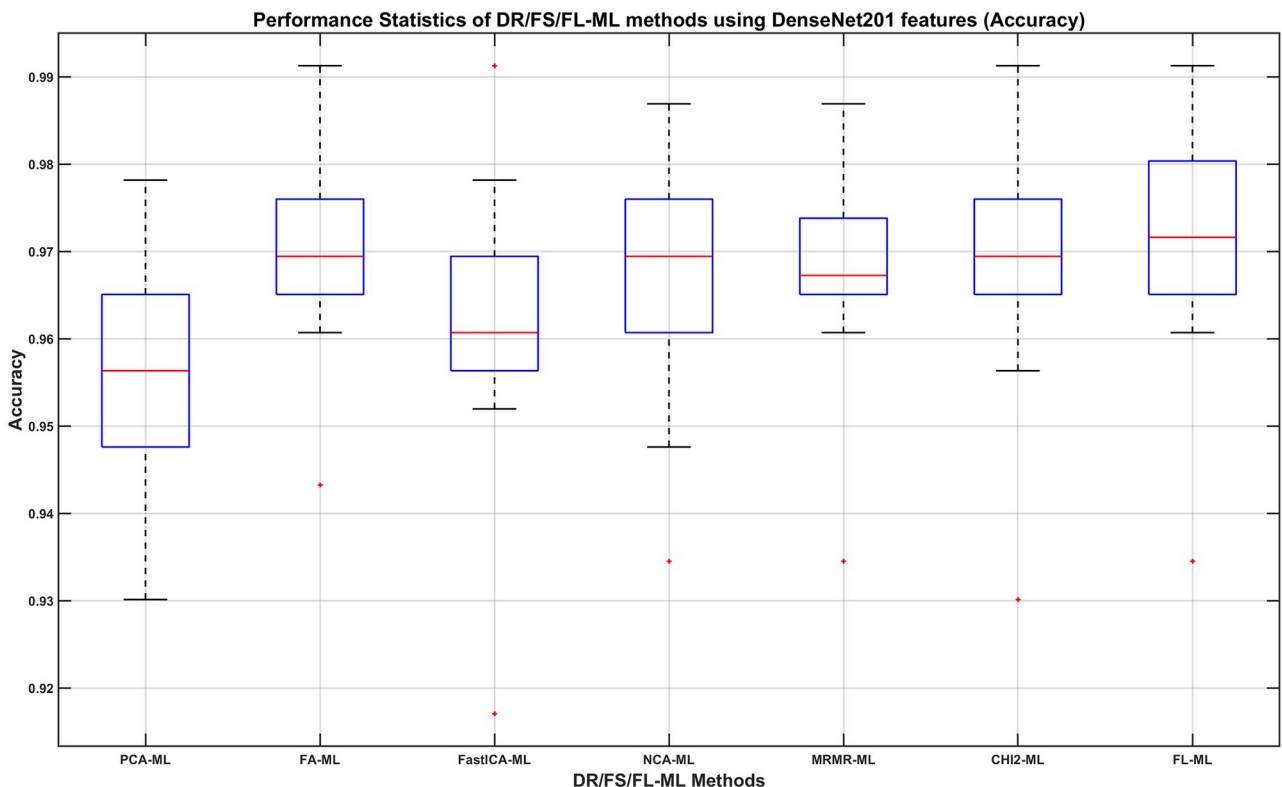

**Fig 8. The Accuracy of DR/FS/FL-ML methods using DenseNet201 features.** The ML learners used can be found in the second row of Table 7.

used. This can be seen from the summary of the classification performance of ML classifiers and FC Neural Networks that are tabulated in Tables 15 and 16 which contain the minimum, the maximum, and the mean values of the performance over the eleven DCNNs provided in Tables 7–10 and 11–14, respectively. Here we can see that the average performance drops significantly from around 0.91 ~ 0.92 (in the case of FL-FC) to 0.80 (in the case of DR-FC) or 0.85 (in the case of FS-FC) whereas there is hardly any difference in average performance between FL-ML, DR-ML, and FS-ML.

By comparing the results in both tables, we conclude that using an ML algorithm is a better option to take than using an FC neural network. Therefore, from the perspective of choosing the best DCNN, we can base our decision only on the ML algorithm classification results. To find out which DCNN that provides the best feature, we tabulate the minimum, maximum, and mean values of the classification performance for each DCNN in Table 17. From the table, we can see that DenseNet201 provides the best performance compared to the other DCNNs as it produces consistently high classification results which range from 0.95 to 0.98 in all performance metrics.

We can then determine, using the DenseNet201 results, which DR/FS method to choose and how its performance compares to using the full-length feature. For this, we show in Figs 8–11, the boxplots of each method's performance calculated over the 20 experiment repeats when the DenseNet201 feature is used. The ML learners that produce the best performance shown in those figures can be found in the second row of Tables 7–10, respectively.

From observing the figures, we conclude that applying a DR or FS method before classification slightly reduces the performance when compared to using the full-length feature,

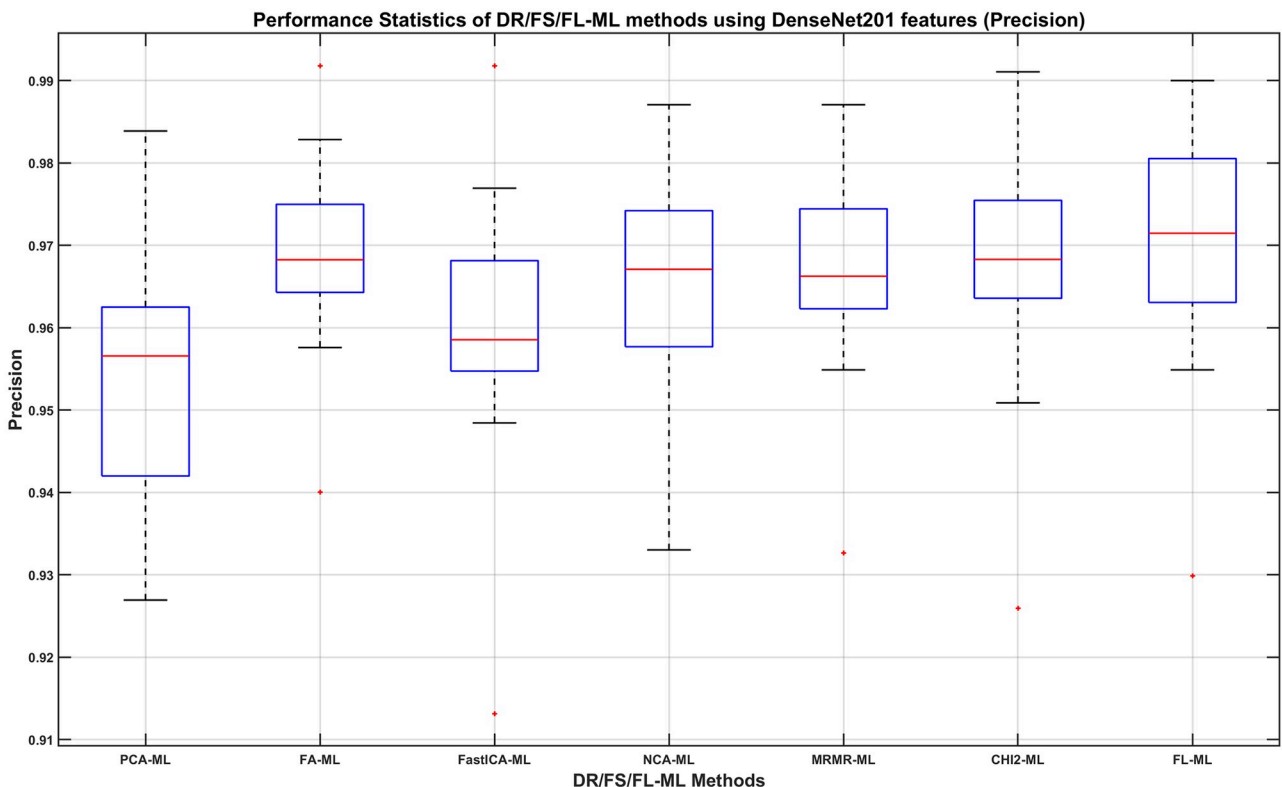

**Fig 9. The Precision of DR/FS/FL-ML methods using DenseNet201 features.** The ML learners used can be found in the second row of Table 8.

although in most cases the differences do not seem to be statistically significant. Given the fact that applying a DR or FS method adds computational cost and time to the process, we concluded that using the full-length feature is the best approach to take. Therefore, we decide that the best combination method is using the full length of the DenseNet201 feature with a Fine Gaussian SVM learner (FL-FGSVM). Using this setup, we can then show the best classification results from the entire set of experiments for each class. These are shown as the per-class classification performance using the Precision, Recall, and F1-score metrics in Figs 12–14.

From Figs 12 and 13, we find that the minimum class precision and recall is around 0.88. The median performance measured using the precision metric ranges from 0.95 to 0.99 whereas that using the recall metric ranges from 0.93 to 1.0. It is worth noting that, in most cases, a computer algorithm is only needed to automatically find the correct L3/L4, L4/L5, and L5/S1 images. In that respect, we can claim that the classification of those three classes when measured using F1-Score, which is the harmonic mean of the precision and recall metrics, is very good. From Fig 14, we can see that the minimum F1-Scores of those three classes range between 0.93 to 0.95, whereas the median F1-Scores range between 0.97 to 0.99.

## 3.4. Consideration on the practical implementation

The whole process we described in this paper has been implemented using MATLAB version 2021a on three different computer setups. One setup has an Intel(R) i7-10700K CPU @ 3.80GHz, 16 GB RAM, and NVIDIA GeForce RTX 3080 with 10 GB VRAM. Another setup has an Intel(R) i7-7700 CPU @ 3.60GHz, 64 GB RAM, and 2x NVIDIA TITAN X with 24 GB

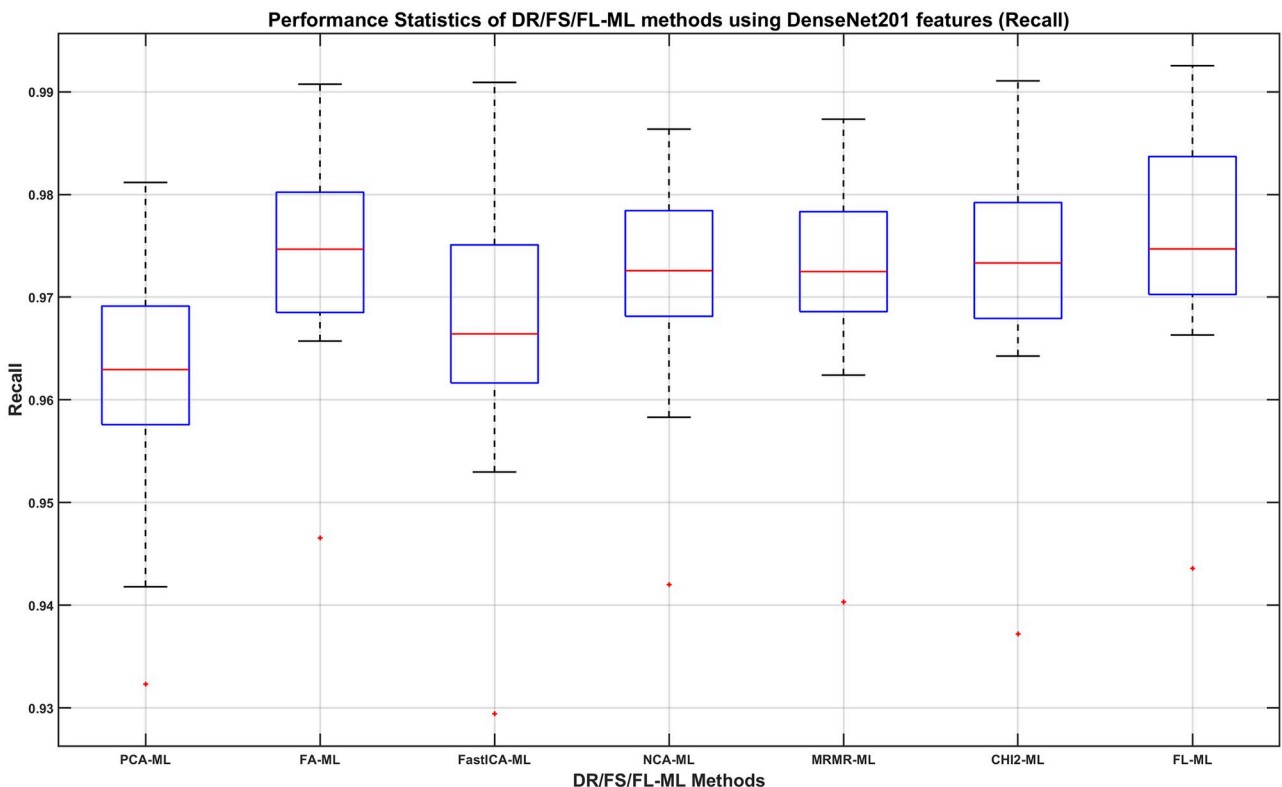

**Fig 10. The Recall of DR/FS/FL-ML methods using DenseNet201 features.** The ML learners used can be found in the second row of Table 9.

VRAM. The last setup has an Intel(R) i9-7900X CPU @ 3.30GHz, 128 GB RAM, and 4x NVI-DIA TITAN XP with 48 GB VRAM. One of the bottlenecks in the experiment is the time taken to train the ML learners and FC neural networks using hyperparameter optimization, which can take hours on the above machines. The source code, dataset, and result files have been made available for review from Mendeley Data [66]. The procedure starts by providing the program with two folders containing identical numbers of T1-weighted and T2-weighted traverse lumbar spine MRI images. The program would assume that the image files in both folders are in the same correct order when sorted alphabetically before applying the image registration step. The image registration results are then stored in another folder. The ground truth information, as a comma-separated value (CSV) file, containing indices of the images that belong to each category is supplied. The program then split the dataset into four folders depending on the CSV file. The program then loads a DCNN and applies it to each image and records the resulting image features. The program also allows a DS or FS algorithm to be applied before using the features for training an ML learner or an FC neural network. The program then uses the trained ML learner or an FC neural network to classify a new image based on the image features. The time taken to extract the feature from an image ranges from 1 to 14 milliseconds and the time to classify one image takes less than 10 milliseconds. The total time would be much faster than manual selection which takes between 30 to 60 seconds, especially if the process is done in big batches.

The same approach can be adopted in a clinic provided that the necessary hardware and software requirements are met, which can be obtained from the MATLAB official website. Some modifications to the source code might be needed to adapt it to each user's setup and

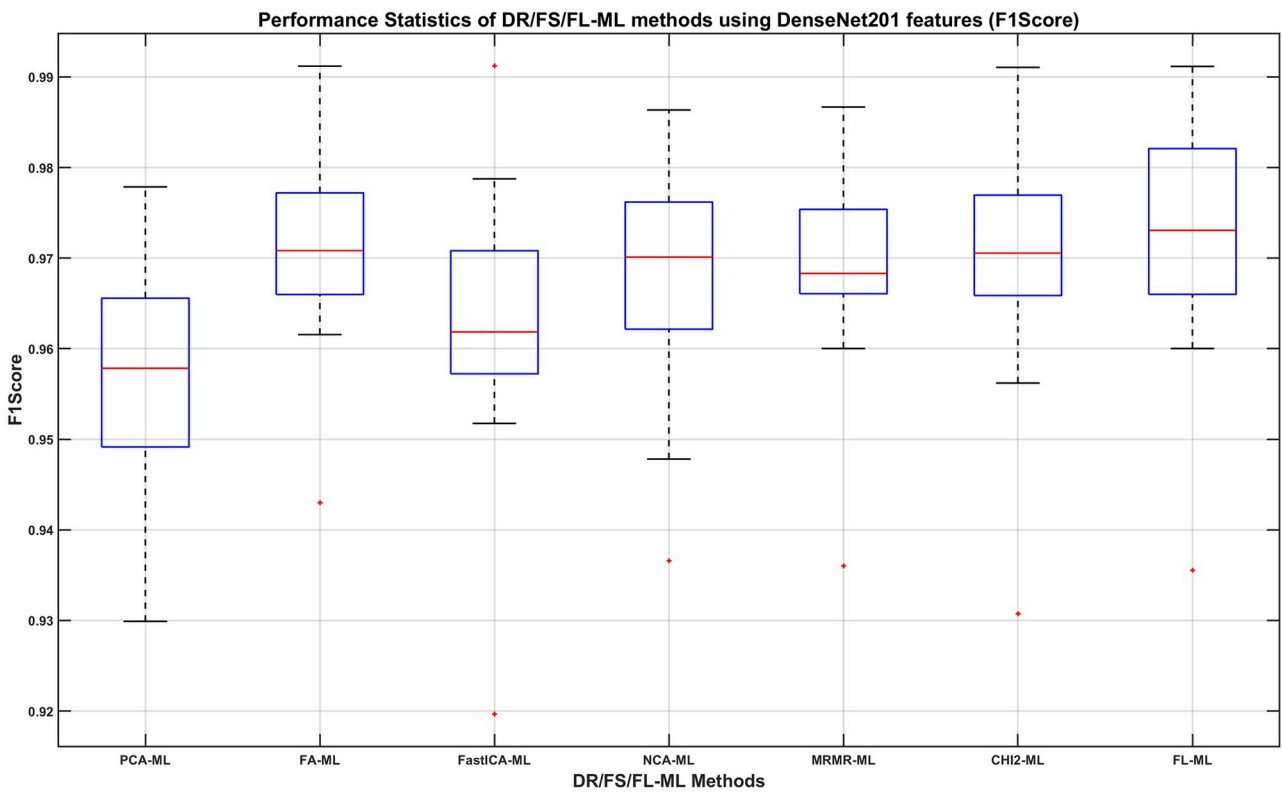

**Fig 11. The F1-Score of DR/FS/FL-ML methods using DenseNet201 features.** The ML learners used can be found in the second row of Table 10.

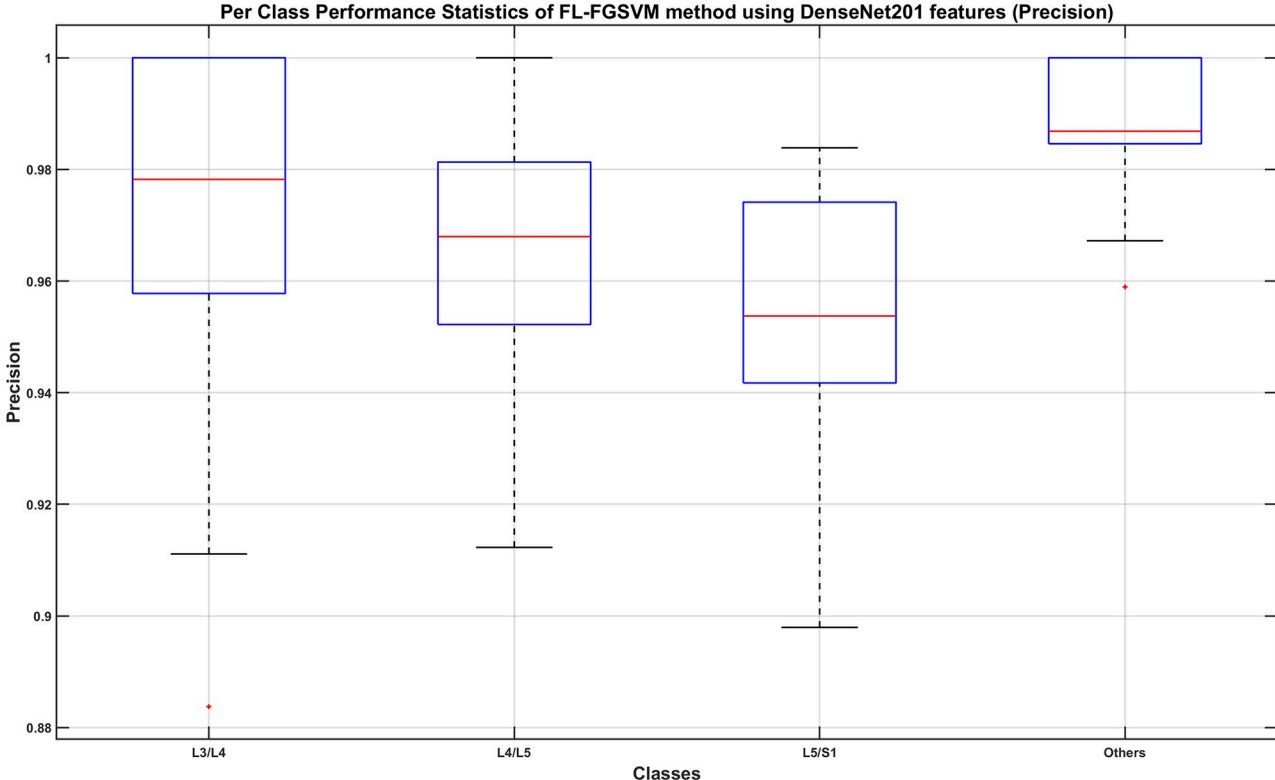

**Fig 12. Per-class classification performance using Fine Gaussian SVM classifier on full-length DenseNet201 features (Precision).**

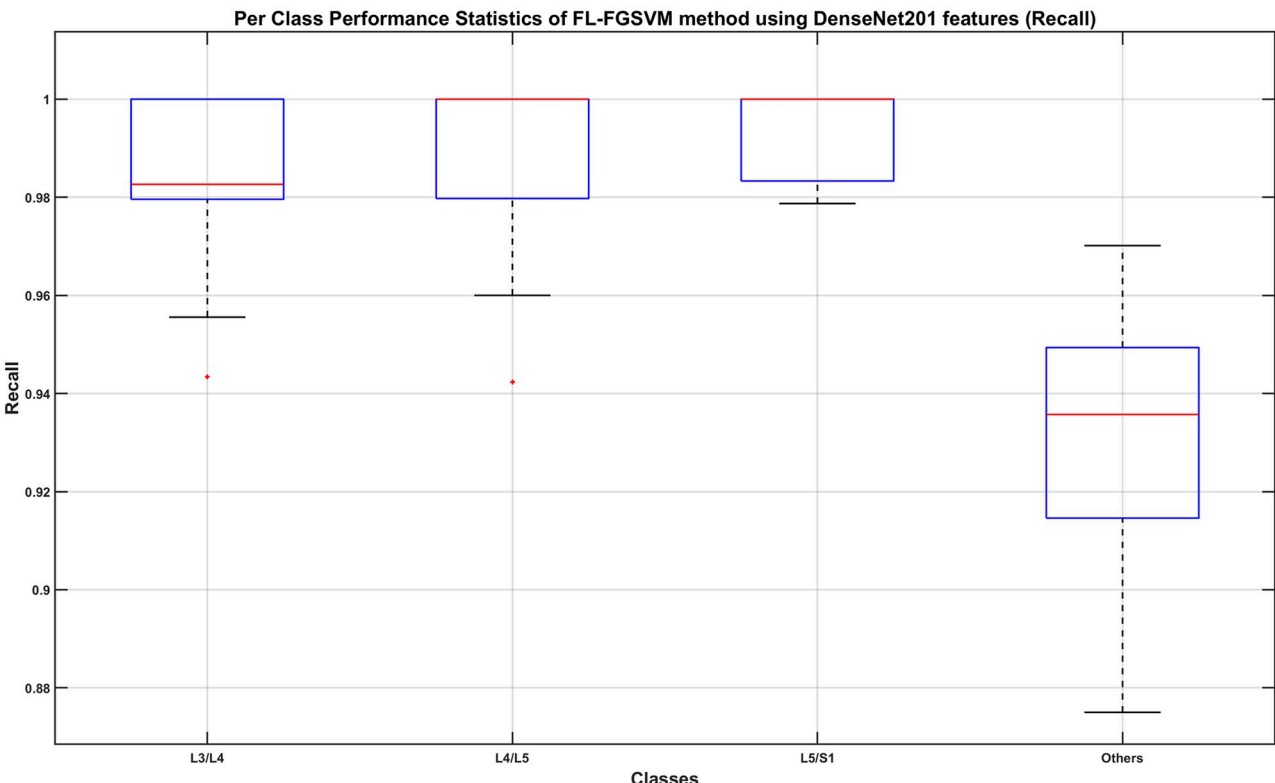

**Fig 13. Per-class classification performance using Fine Gaussian SVM classifier on full-length DenseNet201 features (Recall).**

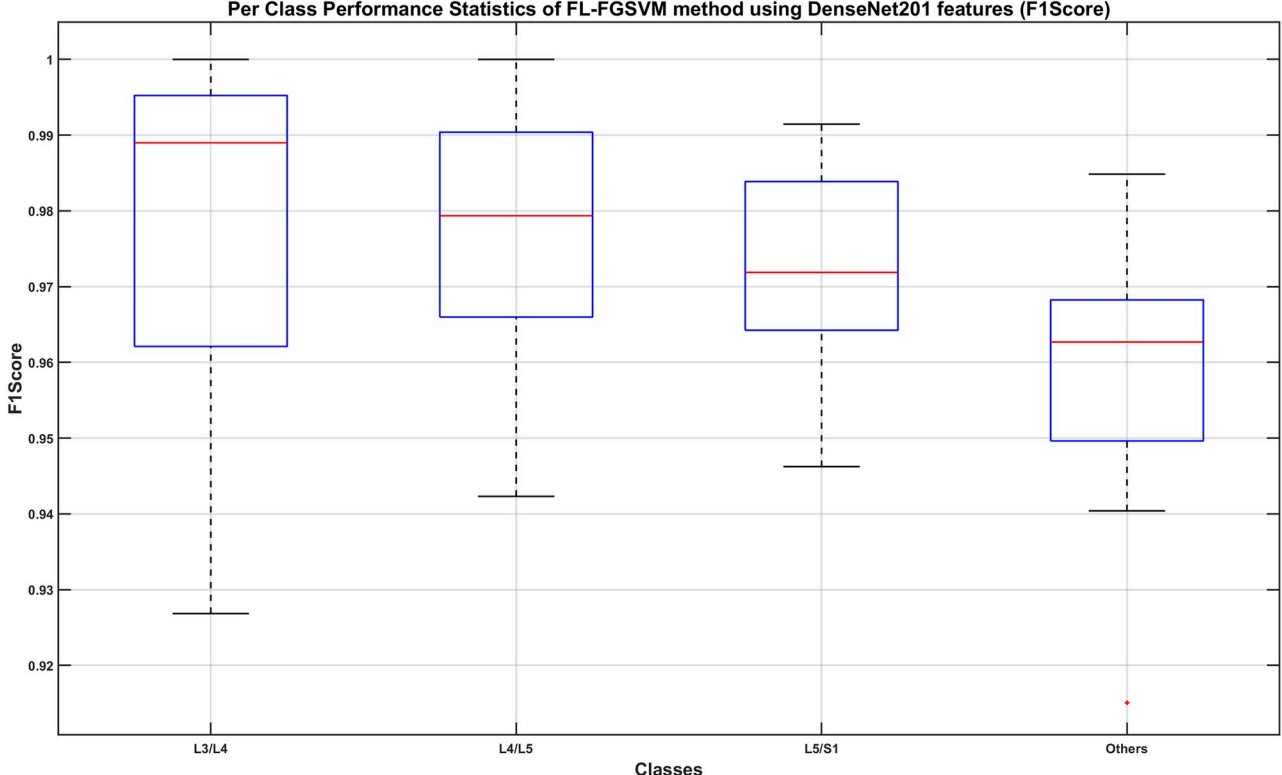

**Fig 14. Per-class classification performance using Fine Gaussian SVM classifier on full-length DenseNet201 features (F1-Score).**

requirement. A similar approach can also be implemented using Python programming language together with the necessary deep learning library (such as Keras [67]) and machine learning library (such as Scikit-Learn [68]). Much of the MATLAB code can be translated to Python but some low-level function implementations could be different.

## 4. Conclusion

We have detailed in this paper, our approach for selecting traverse images that cut closest to the half-height of an Intervertebral Disc from a dataset of traverse lumbar spine MRI. The method is based on using the image features extracted from a Deep Convolutional Neural Network to train a Machine Learning classifier or a Fully Connected Neural Network. We investigated the suitability and usefulness of applying a Dimensionality Reduction technique or a Feature Selection technique prior to the training as well as using the full-length features. In total, we tested eleven Deep Convolutional Neural Network models, three Dimensionality Reduction techniques, three Feature Selection techniques, twenty Machine Learning learners, and three Fully Connected Neural Network learning optimizers. The learners and optimizers were trained using hyperparameter optimization to ensure that they produce the best result they can. We implemented the different method combinations twenty times to get representative results from each method. Our experiment shows that applying the Support Vector Machine algorithm with a short Gaussian kernel on full-length DenseNet201 is the best approach to use. This approach gives the minimum per-class classification performance of around 0.88 when measured using the precision and recall metrics. The median performance measured using the precision metric ranges from 0.95 to 0.99 whereas that using the recall metric ranges from 0.93 to 1.0. When only considering the L3/L4, L4/L5, and L5/S1 classes, the minimum F1-Scores range between 0.93 to 0.95, whereas the median F1-Scores range between 0.97 to 0.99.

### 4.1. Appendix

The MATLAB source files and traverse image subset of the Lumbar Spine MRI Dataset [35] used in this experiment can be downloaded from Mendeley Data [66].

## Supporting information

**S1 Data. A Word document containing URL of the dataset.** Models and PYTHON/ MATLAB source code to reproduce the results.
(DOCX)

## Author Contributions

**Conceptualization:** Sud Sudirman.

**Funding acquisition:** Friska Natalia.

**Investigation:** Friska Natalia, Reyhan Eddy Yunus.

**Methodology:** Friska Natalia, Sud Sudirman.

**Project administration:** Friska Natalia.

**Resources:** Hira Meidia.

**Software:** Julio Christian Young, Nunik Afriliana.

**Supervision:** Sud Sudirman.

**Validation:** Reyhan Eddy Yunus.

**Writing – review & editing:** Friska Natalia, Sud Sudirman.

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
