## [Decision Letter · Decision Letter 0]

11 Apr 2021

PONE-D-21-04349

Automated Selection of Mid-Height Intervertebral Disc Slice in Traverse Lumbar Spine MRI using Transfer Learning and Dimensionality Reduction of Pre-trained DCNN Features

PLOS ONE

Dear Dr. Sudirman,

Thank you for submitting your manuscript to PLOS ONE. After careful consideration, we feel that it has merit but does not fully meet PLOS ONE’s publication criteria as it currently stands. Therefore, we invite you to submit a revised version of the manuscript that addresses the points raised during the review process.

We look forward to receiving your revised manuscript.

Kind regards,

Khanh N.Q. Le

Academic Editor

PLOS ONE

Journal Requirements:

We note that you have stated that you will provide repository information for your data at acceptance. Should your manuscript be accepted for publication, we will hold it until you provide the relevant accession numbers or DOIs necessary to access your data. If you wish to make changes to your Data Availability statement, please describe these changes in your cover letter and we will update your Data Availability statement to reflect the information you provide.

Reviewers' comments:

Reviewer's Responses to Questions

**Comments to the Author**

1. Is the manuscript technically sound, and do the data support the conclusions?

Reviewer #1: No

Reviewer #2: Yes

Reviewer #3: Partly

Reviewer #4: Partly

2. Has the statistical analysis been performed appropriately and rigorously? 

Reviewer #1: Yes

Reviewer #2: Yes

Reviewer #3: Yes

Reviewer #4: Yes

3. Have the authors made all data underlying the findings in their manuscript fully available?

Reviewer #1: No

Reviewer #2: Yes

Reviewer #3: Yes

Reviewer #4: Yes

4. Is the manuscript presented in an intelligible fashion and written in standard English?

Reviewer #1: No

Reviewer #2: Yes

Reviewer #3: Yes

Reviewer #4: Yes

5. Review Comments to the Author

Reviewer #1: The technical detail of this work is missing. Authors should add the technical detail related to proposed method. Without technical detail, the manuscript is not acceptable. Also, based on just results, it is not possible to accept this version.

Reviewer #2: This paper still needs improvement before acceptance for publication. My detailed comments and suggestions are given as follows:

1. The Introduction needs to be revised and should emphasize the challenges and corresponding techniques.

2. More comparative experiments should be added to illustrate the superiority of the propose method, and the experimental results should be further analyzed.

3. Training details should be presented, such as the setting of learning rate and the decay of loss function.

4. Discussions about the generalization performance of deep learning model are encouraged.

5. Many figures are so blurred that they cannot be read, such as Fig. 11, Fig .12, and Fig. 13.

Reviewer #3: In this paper, the authors tackled an interesting problem of selecting specific slices from lumbar spine MRI. The motivation behind this work is clear, and the ideas presented in this manuscript are valid, but it suffers from the following shortcomings that need to be addressed before it could be considered for publication:

1. The authors sort of failed to contextualize their work within the state of the art, as reducing the dimensionality of deep features extracted using deep models is not novel. As the examples, see the following works: https://www.sciencedirect.com/science/article/abs/pii/S1047320319301932, https://link.springer.com/chapter/10.1007/978-3-319-77538-8_34. The authors should not only discuss such techniques in the related literature part of the manuscript, but should also confront their dimensionality reduction technique with other feature extraction and selection algorithms.

2. I suggest removing acronyms from the title.

3. Overall, the quality of the figures is very low and should be substantially improved.

4. The authors should discuss the models presented in Table 1 in much more detail. Specifically, I encourage the authors to prepare a taxonomy of investigated deep architectures (with special emphasis put on their architectural choices that are specific). To this end, the authors should make sure that the manuscript is self-contained. Also, please add the year for each model in Table 1.

5. Please discuss the co-registration process in more detail (lines starting from 296).

6. The authors should perform rigorous crossvalidation to fully understand the generalization abilities of the algorithms (a single 80/20 split may be not enough to infer correct conclusions).

7. Although the authors did try to show different experimental aspects of various architectures, the experiments are rather not thorough. It would be best to present the ablation study (e.g., selection of optimizers) for a wider range of investigated models. Overall, the authors should rework their experimental part of the paper to make it more thorough.

Reviewer #4: Review Manuscript PONE-D-21-04349

The authors present an alternative methodology to assess the selection of mid-high IVD image slide from MRI acquisitions.

This is an interesting work for the spine community since it states a possible automatic way the selection of suitable images of the IVD in lumbar spine. The use deep learning algorithm to a large number of images to test their technique. This work presents a really nice use of deep learning to tackle a difficulty of obtaining better information of the lumbar IVDs. However, the manuscript summited is not ready to be published since there are some aspects to be considered. These aspects are listed as follow:

- Most of the research articles follow the structure: Introduction, Materials and methods, Results, Discussion, Conclusions (sometimes included at the end of the discussion section). This structure is not totally followed by the authors making difficult the follow up of the paper.

- I really enjoyed the introduction part, it was really instructive and easy to follow. Nevertheless, a clear aim of the study is missing. At lines 91-92 the mentioned “In this paper, we detailed our approach to automatically …” but if this is an objective is vague, they need to specify the objective(s) of their study.

- The second section can be reduced and included in the introduction that also need to be reduced. In general, the introduction should not be larger than 2 pages, but some exception are in order when the paper is a review which is not the case.

- Material and methods section: it is poor. In this section the authors should present the steps they did in the study, a description of the methods, i.e. what they did, what they use, what they modified. The database use, the test they did and what were the variable measured/evaluated, they will compare their results with what database to validate.

- Part 4, is a mix of several part. Here the authors present, part of the methods, then present the results, and discuss a little bit about the results. I highly suggest to the authors to follow the article structure previous mentioned.

- Results: in general, they are interesting. They should have a separate section where they can be presented in a proper way

- Discussion: The results are poorly discussed. The authors make some interesting comments regarding the results. However, a comparison with other studies, what are the similarities and differences, and the explanation of the differences, what are the limitations of the method propose, are missed in the manuscript presented.

- Conclusions: is weak. It is not clear the contribution of the new technique to the spine community, does the outcomes obtained are better than the one obtained manually? Is it faster? Can be implemented easily in clinic?

- Figures with bars: It might be better to present a feature with the modes used instead of having them separated. I mean, for example figure 6, the metric accuracy can have the bars for SGDM and Adam, as such, it can be seen better the differences between the two models used.

- Tables 2 and 3: please rearrange the information to better understand the content, also when present statistical results avoid to use scientific notation for the numbers, it is better and more easy to evaluate the number with decimals.

- Anachronisms, check that all of them are previously introduced.

-Experiments: the word “experiment” is most common for cells, object, assay experiments. It is better to use test when you use numerical “experiment”, e.g. testing a new numerical algorithm against another one to evaluate its performance.

- Lines 527 -528: the authors present that their method improves upon the benchmark TL/FT methods by presenting increments in the mean values. How this is true? I mean, how the fact that the mean is higher, is a sign of improvement?

The manuscript is well written and the study is really interesting for spine community. The reviewer encourages the authors to consider all the comments mentioned previously. With the changes suggested in this review, the authors can resubmit the manuscript for publication.

6. PLOS authors have the option to publish the peer review history of their article (what does this mean?). If published, this will include your full peer review and any attached files.

Reviewer #1: No

Reviewer #2: No

Reviewer #3: No

Reviewer #4: No

---

## [Author Response · Author response to Decision Letter 0]

2 Jul 2021

Dear reviewers,

Thank you for all your effort and help in improving our paper. We appreciate your comments, suggestions, and critique. Based on your feedback, we have reworked much of our methodology and experimentation. Our approach was previously proposed using one specific combination of Deep Convolutional Neural Networks and one Dimensionality Reduction to perform the image classification. Our approach now is to test and compare many different combinations of Deep Convolutional Neural Networks for feature extraction, Dimensionality Reduction as well as Feature Selection techniques, several Machine Learning algorithms, several learning optimizers for Fully Connected Neural Networks, and use hyperparameter optimization to get the best classifier for each combination. 

As a result, we pretty much changed over 95% of the texts in the paper.

Furthermore, the time taken to conduct the experiment had significantly increased hence the long delay in resubmitting this revision.

I hope you will find the new version of our paper much better than the last version.

Kind regards

Sud

 

Reviewer #1: The technical detail of this work is missing. Authors should add the technical detail related to proposed method. Without technical detail, the manuscript is not acceptable. Also, based on just results, it is not possible to accept this version.

Thank you for your feedback. The requested technical detail has been added to the paper by expanding the Material section and adding a table. The added information reads:

The material used in this research is taken from our Lumbar Spine MRI Dataset which is available publicly [6,23]. This dataset contains anonymized clinical MRI studies of 515 patients with symptomatic back pains. The dataset consists of 48,345 T1-weighted and T2-weighted traverse and sagittal images of the patients’ lumbar spine in the Digital Imaging and Communications in Medicine (DICOM) format. The images were taken using a 1.5-Tesla Siemens Magnetom Essenza MRI scanner. Most of the images were taken when the patients were in the Head-First-Supine position, though a few were taken when they were in the Feet-First-Supine position. The duration of each patient study ranges between 15 to 45 minutes with time gaps between taking the T1- and T2-weighted scans ranging between 1 to 9 minutes. The patient might have made some movements between the T1 and T2 recordings, which suggests that corresponding T1- and T2- slices may not necessarily align and may require an application of an image registration algorithm to align them. The scanning sequence used in all scans is Spin Echo (SE), which is produced by pairs of radiofrequency pulses, with segmented k-space (SK), spoiled (SP), and oversampling phase (OSP) sequence variant. Fat-Sat pulses were applied just before the start of each imaging sequence to saturate the signal from fat matters to make it appear distinct to water. The summary of the technical information of the scanning parameters carried out when recording these images is provided in Table 1.

Reviewer #2: This paper still needs improvement before acceptance for publication. My detailed comments and suggestions are given as follows:

1. The Introduction needs to be revised and should emphasize the challenges and corresponding techniques.

Thank you for your comments. The Introduction section has been revised and the challenges of the problem and the corresponding techniques have been added. The specific part of the section that addresses this now reads:

Based on the above argument, we believe that both a) the lack of directly relevant methods proposed in the literature that selects the best traverse plane that cuts closest to the half-height of an IVD in a lumbar spine MRI and b) the wide range of potentially suitable DR or FS methods and image classification methods, provide the rationale and urgency for this study. The aim of this study is to find the best method to select the best traverse plane that cuts closest to the half-height of an IVD in a lumbar spine MRI by studying and comparing the different combination of machine learning methods and approaches. We report the result of our investigation on the suitability and performance of different approaches of machine learning in solving the aforementioned medical image classification challenge. The contributions of this work are summarized as follows:

 Investigated the classification performance using image features calculated using eleven different pre-trained DCNN models.

 Investigated the effect of three dimensionality-reduction techniques and three feature-selection techniques on the classification performance.

 Investigated the performance of five different ML algorithms and three FC learning optimizers which are trained with hyperparameter optimization using a wide range of hyperparameter options and values.

2. More comparative experiments should be added to illustrate the superiority of the propose method, and the experimental results should be further analyzed.

We have reworked the methodology by using and comparing a) the image features of eleven Deep Convolution Neural Network architectures, b) Three Dimensionality Reduction techniques (PCA, FA and Fast ICA) and three Feature Selection techniques (NCA, MRMR and CHI2), and c) Five different Machine Learning algorithms and three Fully Connected neural Network learning optimizers. The results have been comparatively analyzed in the Experimental Results and Analysis section. The best method, which was found to be using full length DenseNet201 features with Support Vector Machine algorithm and a short Gaussian kernel, was found by comparing the results of all combinations.

3. Training details should be presented, such as the setting of learning rate and the decay of loss function.

The training details have been presented in Table 4 and Table 5. These include all the fixed and variables and range of values used during the hyperparameter optimization. Due to the large number of models trained it is very impractical to add present how the loss values decay during training.

4. Discussions about the generalization performance of deep learning model are encouraged.

We conducted the experiment 20 times, each using different training and test sets, although the ratio remains the same (which is 80:20). The generalization of the performance is discussed by presenting and analyzing the statistics of the results including the minimum, maximum and median values of the performance metrics used (line 463-559).

5. Many figures are so blurred that they cannot be read, such as Fig. 11, Fig .12, and Fig. 13.

All the figures have been redone as part of the rework of the experimentation. 

Reviewer #3: In this paper, the authors tackled an interesting problem of selecting specific slices from lumbar spine MRI. The motivation behind this work is clear, and the ideas presented in this manuscript are valid, but it suffers from the following shortcomings that need to be addressed before it could be considered for publication:

1. The authors sort of failed to contextualize their work within the state of the art, as reducing the dimensionality of deep features extracted using deep models is not novel. As the examples, see the following works: https://hes32-ctp.trendmicro.com:443/wis/clicktime/v1/query?url=https%3a%2f%2fwww.sciencedirect.com%2fscience%2farticle%2fabs%2fpii%2fS1047320319301932&umid=2356ab6a-0087-4c78-a443-85e54e732539&auth=768f192bba830b801fed4f40fb360f4d1374fa7c-1e228c95a24c231c547f34926f2ce63065296b67, https://hes32-ctp.trendmicro.com:443/wis/clicktime/v1/query?url=https%3a%2f%2flink.springer.com%2fchapter%2f10.1007%2f978%2d3%2d319%2d77538%2d8%5f34&umid=2356ab6a-0087-4c78-a443-85e54e732539&auth=768f192bba830b801fed4f40fb360f4d1374fa7c-6295542040e70308207ac0bfc066063669ba3bd9. The authors should not only discuss such techniques in the related literature part of the manuscript, but should also confront their dimensionality reduction technique with other feature extraction and selection algorithms.

Thank you for your advice. 

We have reworked the methodology by using and comparing a) the image features of eleven Deep Convolution Neural Network architectures, b) Three Dimensionality Reduction techniques (PCA, FA and Fast ICA) and three Feature Selection techniques (NCA, MRMR and CHI2), and c) Five different Machine Learning algorithms and three Fully Connected neural Network learning optimizers. The results have been comparatively analyzed in the Experimental Results and Analysis section. In the new version of the paper, we emphasize on the comprehensiveness of the combination of methods that we implemented and the approach that we took to get the best overall method combination from them. The best method, which was found to be using full length DenseNet201 features with Support Vector Machine algorithm and a short Gaussian kernel, was found by comparing the results of all combinations.

2. I suggest removing acronyms from the title.

All acronyms have been removed from the title and the title has been replaced to reflect the new methodology better.

3. Overall, the quality of the figures is very low and should be substantially improved.

All the figures have been redone as part of the rework of the experimentation.

4. The authors should discuss the models presented in Table 1 in much more detail. Specifically, I encourage the authors to prepare a taxonomy of investigated deep architectures (with special emphasis put on their architectural choices that are specific). To this end, the authors should make sure that the manuscript is self-contained. Also, please add the year for each model in Table 1.

We could not find any resources in the literature that provides a good taxonomy of DCNN architecture. Hence, it is difficult for us to provide one. However, we added a short taxonomy of deep neural network architectures in general that reads:

DCNN is one of several types of deep neural networks that gain popularity in recent years to solve many artificial intelligence problems. Different types of deep neural networks have significantly different architectures and are designed to solve different types of problems. DCNNs are typically used for image classification. Recurrent Neural Networks, such as Long Short-Term Memory [21], are used to recognize patterns in sequences of data such as time-series data, speech, and texts. There are also Fully Convolutional Neural Networks, such as U-net [22] and SegNet [23] that are used mainly for semantic image segmentation. Some DCNNs have also been modified to become Region-based CNNs [24,25] to detect and recognize multiple objects within an image.

In the previous version, we only investigated two deep convolutional neural network (DCNN) architectures. Now we have used eleven architectures. We have added the year information to the summary of the architectures used as shown in Table 3. 

5. Please discuss the co-registration process in more detail (lines starting from 296).

More information on the image registration process and examples of the results have been added. The texts now become:

From each pair of T1-weighted and T2-weighted images, a 3-channel composite image is created resulting in 8,936 composite images. The first channel of the composite image is constructed from the T1-weighted image, the second channel is constructed from the image-registered T2-weighted image, and the last channel is constructed from the Manhattan distance of the two. The image used to construct the second channel is obtained by performing image registration on the T2-weighted image to its T1-weighted counterpart to ensure that every pixel at the same location in both images corresponds to the same voxel in an organ or tissue. This is performed by finding the minimum difference between the fixed T1-weighted image and a set of transformed T2-weighted images calculated over a search space of affine transforms. Mathematically, the process can be described as follows: Let I_R (v) be the reference 2D image and I_T (v) be the to-be-transformed 2D image, where v=〖[x,y,z]〗^Tis a real-valued voxel location. The voxel location v is defined on the continuous domains V_R and V_T, that corresponds to each pixel in I_R and I_T, respectively. Note that in our case, I_R and I_T are the T1-weighted image and the T2-weighted image, respectively. The image registration that we employ in this method is a process that seeks a set of transformation parameters μ ^ from all sets of transformation parameters μ that minimizes the image discrepancy function S

 μ ^=(arg min)┬μ⁡〖S(I_R,I_T∘g(v│μ))〗 (1)

We calculate S using Matte’s mutual information metric described in [29] over a search space in μ domain. The search process uses an iterative process called the Evolutionary Algorithm that perturbs, or mutates, the parameters from the last iteration. If the new perturbed parameters yield a better result than the last iteration, then more perturbation is applied to the parameters in the next iteration, otherwise a less aggressive perturbation is applied. The search process is optimized using the (1+1)-Evolutionary Strategy [30] which locally adjusts the search direction and step size and provides a mechanism to step out of non-optimal local minima. The search is carried out up to 300 iterations with a parameter growth factor of 1.05. A sequence of parametric bias field estimation and correction method, called PABIC [30], is applied to counter the effect of low-frequency inhomogeneity field and high-frequency noise on both T1 and T2 modalities.

There are several cases where the image registration process fails because the algorithm is unable to converge. This could be because the patient’s position and orientation when the two scans were recorded differ significantly. In this case, the images were removed from the dataset. We show in Figure 4, two example cases where the image registration process succeeded (left column) and failed (right column).

Figure 4. Two example cases where the image registration process succeeded (left column) and failed (right column). The top row shows the T1-weighted images, the middle row shows T2-weighted images, and the bottom row shows the resulting composite images after image registration.

6. The authors should perform rigorous crossvalidation to fully understand the generalization abilities of the algorithms (a single 80/20 split may be not enough to infer correct conclusions).

We have reworked the methodology by using and comparing a) the image features of eleven Deep Convolution Neural Network architectures, b) Three Dimensionality Reduction techniques (PCA, FA and Fast ICA) and three Feature Selection techniques (NCA, MRMR and CHI2), and c) Five different Machine Learning algorithms and three Fully Connected neural Network learning optimizers. The results have been comparatively analyzed in the Experimental Results and Analysis section.

We conducted the experiment 20 times, each using different training and test sets, although the ratio remains the same (which is 80:20). During training, a smaller subset of the training set is allocated as the validation set to measure the classifier’s performance during training. The generalization of the performance is discussed by presenting and analyzing the statistics of the results including the minimum, maximum and median values of the performance metrics used (line 450-546).

7. Although the authors did try to show different experimental aspects of various architectures, the experiments are rather not thorough. It would be best to present the ablation study (e.g., selection of optimizers) for a wider range of investigated models. Overall, the authors should rework their experimental part of the paper to make it more thorough.

In this new version of the paper, the Machine Learning algorithms have been trained with hyperparameter optimization using a wide range of hyperparameter options and values. Similarly, the Fully Connected Neural Networks have been trained using three different learning optimizers with a wide range of hyperparameter options and values. The list of hyperparameter optimization values, range of values and options is given in Table 4 and 5. 

Reviewer #4: Review Manuscript PONE-D-21-04349

The authors present an alternative methodology to assess the selection of mid-high IVD image slide from MRI acquisitions.

This is an interesting work for the spine community since it states a possible automatic way the selection of suitable images of the IVD in lumbar spine. The use deep learning algorithm to a large number of images to test their technique. This work presents a really nice use of deep learning to tackle a difficulty of obtaining better information of the lumbar IVDs. However, the manuscript summited is not ready to be published since there are some aspects to be considered. These aspects are listed as follow:

- Most of the research articles follow the structure: Introduction, Materials and methods, Results, Discussion, Conclusions (sometimes included at the end of the discussion section). This structure is not totally followed by the authors making difficult the follow up of the paper.

Thank you for your feedback.

We have restructured the organization of the paper to follow your suggestion. There are now Introduction section, Material and Method section, Experimental Results, Analysis and Discussion section, and lastly the Conclusion section. We opted to combine the presentation of experimental results, with their analysis and the discussion into one section to make it easier to discuss the many different results and points.

- I really enjoyed the introduction part, it was really instructive and easy to follow. Nevertheless, a clear aim of the study is missing. At lines 91-92 the mentioned “In this paper, we detailed our approach to automatically …” but if this is an objective is vague, they need to specify the objective(s) of their study.

We have reworded much of the Introduction section to provide the rationale of the study. In addition, a summary of the study rationale, aim and contribution is given at the end of the section which reads:

Based on the above argument, we believe that both a) the lack of directly relevant methods proposed in the literature that selects the best traverse plane that cuts closest to the half-height of an IVD in a lumbar spine MRI and b) the wide range of potentially suitable DR or FS methods and image classification methods, provide the rationale and urgency for this study. The aim of this study is to find the best method to select the best traverse plane that cuts closest to the half-height of an IVD in a lumbar spine MRI by studying and comparing the different combination of machine learning methods and approaches. We report the result of our investigation on the suitability and performance of different approaches of machine learning in solving the aforementioned medical image classification challenge. The contributions of this work are summarized as follows:

a) Investigated the classification performance using image features calculated using eleven different pre-trained DCNN models.

b) Investigated the effect of three dimensionality-reduction techniques and three feature-selection techniques on the classification performance.

c) Investigated the performance of five different ML algorithms and three FC learning optimizers which are trained with hyperparameter optimization using a wide range of hyperparameter options and values.

- The second section can be reduced and included in the introduction that also need to be reduced. In general, the introduction should not be larger than 2 pages, but some exception are in order when the paper is a review which is not the case.

The two sections have been combined and reduced in length. However, it is still not of two pages that you recommended this is due to we use the section to provide a) background information, b) relevant technologies and techniques (since we no longer have the related work section), c) rationale for the study, and d) adding additional information requested by other reviewers.

- Material and methods section: it is poor. In this section the authors should present the steps they did in the study, a description of the methods, i.e. what they did, what they use, what they modified. The database use, the test they did and what were the variable measured/evaluated, they will compare their results with what database to validate.

We have reworked the Material and Method section which now contains more detailed information, including technical information about the dataset and the methodology used. We used a standard approach in machine learning by training and testing the classifiers using a mutually exclusive sets called training and test sets, respectively. The sets are determined randomly from the entire dataset. The experiment is repeated 20 times, each using different training and test sets, although the ratio remains the same (which is 80:20).

Since the classifiers are tested using a different set of images for its training, we can infer on the generality of the classifier by analyzing its performance. 

- Part 4, is a mix of several part. Here the authors present, part of the methods, then present the results, and discuss a little bit about the results. I highly suggest to the authors to follow the article structure previous mentioned.

We have now separated the methodology from the experiment result analysis and discussion. The two sections now will read very distinctly.

- Results: in general, they are interesting. They should have a separate section where they can be presented in a proper way

- Discussion: The results are poorly discussed. The authors make some interesting comments regarding the results. However, a comparison with other studies, what are the similarities and differences, and the explanation of the differences, what are the limitations of the method propose, are missed in the manuscript presented.

We have one section that present the results, their analysis and the discussion. We have tried before to separate the three but due to the large number of results to present, as well as the interconnection between one sets of results to the next, we feel that doing so will make the paper even hard to read. So, we decided to put them into one section so we can present, analyze and discuss each set of results in order.

- Conclusions: is weak. It is not clear the contribution of the new technique to the spine community, does the outcomes obtained are better than the one obtained manually? Is it faster? Can be implemented easily in clinic?

We have added a subsection in section 3 called “Consideration on the practical implementation” where we discussed the above points. the subsection reads:

3.4 Consideration on the practical implementation

The whole process we described in this paper has been implemented using MATLAB version 2021a on three different computer setups. One setup has an Intel(R) i7-10700K CPU @ 3.80GHz, 16 GB RAM, and NVIDIA GeForce RTX 3080 with 10 GB VRAM. Another setup has an Intel(R) i7-7700 CPU @ 3.60GHz, 64 GB RAM, and 2x NVIDIA TITAN X with 24 GB VRAM. The last setup has an Intel(R) i9-7900X CPU @ 3.30GHz, 128 GB RAM, and 4x NVIDIA TITAN XP with 48 GB VRAM. One of the bottlenecks in the experiment is the time taken to train the ML learners and FC neural networks using hyperparameter optimization, which can take hours on the above machines. The source code, dataset, and result files have been made available for review from Mendeley Data [59]. The procedure starts by providing the program with two folders containing identical numbers of T1-weighted and T2-weighted traverse lumbar spine MRI images. The program would assume that the image files in both folders are in the same correct order when sorted alphabetically before applying the image registration step. The image registration results are then stored in another folder. The ground truth information, as a comma-separated value (CSV) file, containing indices of the images that belong to each category is supplied. The program then split the dataset into four folders depending on the CSV file. The program then loads a DCNN and applies it to each image and records the resulting image features. The program also allows a DS or FS algorithm to be applied before using the features for training an ML learner or an FC neural network. The program then uses the trained ML learner or an FC neural network to classify a new image based on the image features. The time taken to extract the feature from an image ranges from 1 to 14 milliseconds and the time to classify one image takes less than 10 milliseconds. The total time would be much faster than manual selection which takes between 30 to 60 seconds, especially if the process is done in big batches.

The same approach can be adopted in a clinic provided that the necessary hardware and software requirements are met, which can be obtained from the MATLAB official website. Some modifications to the source code might be needed to adapt it to each user’s setup and requirement. A similar approach can also be implemented using Python programming language together with the necessary deep learning library (such as Keras [60]) and machine learning library (such as Scikit-Learn [61]). Much of the MATLAB code can be translated to Python but some low-level function implementations could be different.

- Figures with bars: It might be better to present a feature with the modes used instead of having them separated. I mean, for example figure 6, the metric accuracy can have the bars for SGDM and Adam, as such, it can be seen better the differences between the two models used.

These figures are no longer in the paper. The experiment results are now presented with boxplot which is a more standard way to present and compare distribution of values statistically.

- Tables 2 and 3: please rearrange the information to better understand the content, also when present statistical results avoid to use scientific notation for the numbers, it is better and more easy to evaluate the number with decimals.

The paper now uses decimal numbers and no longer contains scientific notation for the numbers.

- Anachronisms, check that all of them are previously introduced.

We have checked and made sure that all of acronyms have been previously introduced before using them.

-Experiments: the word “experiment” is most common for cells, object, assay experiments. It is better to use test when you use numerical “experiment”, e.g. testing a new numerical algorithm against another one to evaluate its performance.

We have made the necessary changes on the paper.

- Lines 527 -528: the authors present that their method improves upon the benchmark TL/FT methods by presenting increments in the mean values. How this is true? I mean, how the fact that the mean is higher, is a sign of improvement?

The values that are presented are accuracy, precision, recall and F1-score performance metrics which range from 0 (the worst) to one (the best). The zero value means none of the images are classified correctly whereas one means all of the images are classified correctly. We conducted the experiment 20 times, to ensure that the results that we have is not based on chance and we presented the mean as well as the spread of the result values (as box plots) so that the reader can get a fuller picture of the results.

The manuscript is well written and the study is really interesting for spine community. The reviewer encourages the authors to consider all the comments mentioned previously. With the changes suggested in this review, the authors can resubmit the manuscript for publication.

Thank you for your feedback. I hope the changes that we made have addressed all your concerns.

---

## [Decision Letter · Decision Letter 1]

2 Aug 2021

PONE-D-21-04349R1

Automated Selection of Mid-Height Intervertebral Disc Slice in Traverse Lumbar Spine MRI using a Combination of Deep Learning Feature and Machine Learning Classifier.

PLOS ONE

Dear Dr. Sudirman,

Thank you for submitting your manuscript to PLOS ONE. After careful consideration, we feel that it has merit but does not fully meet PLOS ONE’s publication criteria as it currently stands. Therefore, we invite you to submit a revised version of the manuscript that addresses the points raised during the review process.

We look forward to receiving your revised manuscript.

Kind regards,

Khanh N.Q. Le

Academic Editor

PLOS ONE

Journal Requirements:

Reviewers' comments:

Reviewer's Responses to Questions

**Comments to the Author**

1. If the authors have adequately addressed your comments raised in a previous round of review and you feel that this manuscript is now acceptable for publication, you may indicate that here to bypass the “Comments to the Author” section, enter your conflict of interest statement in the “Confidential to Editor” section, and submit your "Accept" recommendation.

Reviewer #1: (No Response)

Reviewer #2: All comments have been addressed

Reviewer #3: All comments have been addressed

2. Is the manuscript technically sound, and do the data support the conclusions?

Reviewer #1: Partly

Reviewer #2: No

Reviewer #3: Yes

3. Has the statistical analysis been performed appropriately and rigorously? 

Reviewer #1: No

Reviewer #2: No

Reviewer #3: Yes

4. Have the authors made all data underlying the findings in their manuscript fully available?

Reviewer #1: Yes

Reviewer #2: No

Reviewer #3: (No Response)

5. Is the manuscript presented in an intelligible fashion and written in standard English?

Reviewer #1: Yes

Reviewer #2: Yes

Reviewer #3: Yes

6. Review Comments to the Author

Reviewer #1: 1) "A CAD system can help doctors understand the cause of an illness better by automating some steps in the diagnosis process. In a CAD system that uses medical images, the system applies image analysis algorithms to different types or modalities of medical imaging, such as Magnetic Resonance Imaging (MRI), of the patient."- add reference for this statement. I suggest the following:

- Skin lesion segmentation and multiclass classification using deep learning features and improved moth flame optimization

- Computer Decision Support System for Skin Cancer Localization and Classification

- Multimodal brain tumor classification using deep learning and robust feature selection: A machine learning application for radiologists

2) "In the case of MRI, for example, a CAD system might use the two modalities of MRI, namely the T1-weighted and T2-weighted MRI, which can differently highlight various types of tissues based on their fat and water composition."- add figures of T1, T2, T1W, and Flair. You can take this figure from the following:

- A Decision Support System for Multimodal Brain Tumor Classification using Deep Learning

- Microscopic brain tumor detection and classification using 3D CNN and feature selection architecture

3) "Training DCNN models take a long time hence there exist several pre-trained DCNN models that are readily usable for image classification."- add reference for this statement.

4) "used in many other types of image classification tasks, including medical image classification, through a method called Transfer Learning"- add reference for this statement: I suggest the folowing:

- Attributes based skin lesion detection and recognition: A mask RCNN and transfer learning-based deep learning framework

- A deep neural network and classical features based scheme for objects recognition: an application for machine inspection

5) Add manuscript organization before materials and methods section.

6) What represent Table 1?

Reviewer #2: The revision of opinions is not satisfactory. The overall expression and organization of the paper should be further improved.

Reviewer #3: (No Response)

7. PLOS authors have the option to publish the peer review history of their article (what does this mean?). If published, this will include your full peer review and any attached files.

Reviewer #1: No

Reviewer #2: No

Reviewer #3: No

---

## [Author Response · Author response to Decision Letter 1]

4 Aug 2021

Dear reviewers,

Thank you once again for all your effort and help in improving our paper. We appreciate your comments, suggestions, and critique. Based on your latest feedback, we have made the required changes to the paper. I hope that this latest version of the paper is of your satisfaction.

Kind regards

Sud

 

Reviewer #1: 

1) "A CAD system can help doctors understand the cause of an illness better by automating some steps in the diagnosis process. In a CAD system that uses medical images, the system applies image analysis algorithms to different types or modalities of medical imaging, such as Magnetic Resonance Imaging (MRI), of the patient."- add reference for this statement. I suggest the following:

- Skin lesion segmentation and multiclass classification using deep learning features and improved moth flame optimization

- Computer Decision Support System for Skin Cancer Localization and Classification

- Multimodal brain tumor classification using deep learning and robust feature selection: A machine learning application for radiologists

We have added the relevant references to the paper. That sentence now reads, “A CAD system can help doctors understand the cause of an illness better by automating some steps in the diagnosis process. In a CAD system that uses medical images, the system applies image analysis algorithms to different types or modalities of medical imaging, such as Magnetic Resonance Imaging (MRI), of the patient [1–3].”

The added references are:

1. Khan MA, Sharif M, Akram T, Damaševičius R, Maskeliūnas R. Skin lesion segmentation and multiclass classification using deep learning features and improved moth flame optimization. Diagnostics. 2021;11(5):811. 

2. Khan MA, Akram T, Sharif M, Kadry S, Nam Y. Computer Decision Support System for Skin Cancer Localization and Classification. C Mater Contin. 2021;68(1):1041–64. 

3. Khan MA, Ashraf I, Alhaisoni M, Damaševičius R, Scherer R, Rehman A, Bukhari SAC. Multimodal brain tumor classification using deep learning and robust feature selection: A machine learning application for radiologists. Diagnostics. 2020;10(8):565.

2) "In the case of MRI, for example, a CAD system might use the two modalities of MRI, namely the T1-weighted and T2-weighted MRI, which can differently highlight various types of tissues based on their fat and water composition."- add figures of T1, T2, T1W, and Flair. You can take this figure from the following:

- A Decision Support System for Multimodal Brain Tumor Classification using Deep Learning

- Microscopic brain tumor detection and classification using 3D CNN and feature selection architecture

Unfortunately, the figures in those papers are copyrighted materials and we are not allowed to reproduce them in our paper without the publisher’s permission. In their place, we added two examples taken from our dataset. Since we only used T1- and T2-weighted images hence it is only appropriate that we include these two examples only. In the paper, after that sentence we insert the following sentence: “An example of a T1-weighted and a T2-weighted traverse MRI images of the L3/L4 Intervertebral Disc (IVD) of the same patient are shown in Figure 1.”

Figure 1. A T1-weighted (left) and a T2-weighted (right) traverse MRI images of the L3/L4 Intervertebral Disc of a patient are shown. One marked difference in the two images is the cerebrospinal fluid (CSF) in the spinal canal that appears black on the T1-weighted image but as a brighter region on the T2-weighted image because of its low fat contents.

I hope this additional information is a sufficient substitute to your original suggestion.

3) "Training DCNN models take a long time hence there exist several pre-trained DCNN models that are readily usable for image classification."- add reference for this statement.

We have added the relevant references to the paper. That sentence now reads, “Training DCNN models take a long time hence there exist several pre-trained DCNN models that are readily usable for image classification [29,30]”

The added references are:

29. Kornblith S, Shlens J, Le Q V. Do better imagenet models transfer better? In: Proceedings of the IEEE/CVF Conference on Computer Vision and Pattern Recognition. 2019. p. 2661–71. 

30. Morid MA, Borjali A, Del Fiol G. A scoping review of transfer learning research on medical image analysis using ImageNet. Comput Biol Med. 2020;128:104115.

4) "used in many other types of image classification tasks, including medical image classification, through a method called Transfer Learning"- add reference for this statement: I suggest the folowing:

- Attributes based skin lesion detection and recognition: A mask RCNN and transfer learning-based deep learning framework

- A deep neural network and classical features based scheme for objects recognition: an application for machine inspection

We have added the relevant references to the paper. The complete sentence now reads, “However, despite being extracted using a model trained using photographic images, these learnable features are sufficiently general that they can be used in many other types of image classification tasks, including medical image classification, through a method called Transfer Learning [32,33], which process is elucidated in Figure 4.”

The added references are:

32. Khan MA, Akram T, Zhang Y-D, Sharif M. Attributes based skin lesion detection and recognition: A mask RCNN and transfer learning-based deep learning framework. Pattern Recognit Lett. 2021;143:58–66. 

33. Hussain N, Khan MA, Sharif M, Khan SA, Albesher AA, Saba T, Armaghan A. A deep neural network and classical features based scheme for objects recognition: an application for machine inspection. Multimed Tools Appl. 2020;1–23.

5) Add manuscript organization before materials and methods section.

We have added paper organization before materials and methods section that reads: “The organization of this paper is as follows. Section 2 describes the dataset used in the research and the proposed method. The experimental results, analysis, and discussion are discussed in detail in Section 3. We then provide the conclusion of our findings in the last section of the paper.”

6) What represent Table 1?

Table 1 shows the range of acquisition parameter values used during traverse MRI scan. The acquisition parameters are a set of values that the radiologist or technician used when the MRI images are being recorded. The values can be fixed (e.g., in case of Field of View, Matrix, Imaging Frequency, and Flip Angle) or differ from one patient to another depending on the decision made by the technician at that time. This information is extracted from the DICOM images from the dataset and description of each parameter can be found on https://dicom.innolitics.com/ciods/ct-performed-procedure-protocol/performed-ct-reconstruction/00189934/00180050

We have altered the caption of the table slightly to improve the description of the table. It now reads,

Table 1. The range of acquisition parameter values used during traverse MRI scans

Also, for your information, Table 1 is included in the paper because in the previous round of review one of the reviewers required us to include the technical detail of the dataset.

Thank you for your feedback. I hope al the changes that we made above have addressed all your concerns. 

Reviewer #2: The revision of opinions is not satisfactory. The overall expression and organization of the paper should be further improved.

Thank you for your feedback. I hope the changes that we made as part of the review from the other reviewer have addressed all your concerns. 

Reviewer #3: (No Response)

---

## [Decision Letter · Decision Letter 2]

4 Oct 2021

PONE-D-21-04349R2Automated Selection of Mid-Height Intervertebral Disc Slice in Traverse Lumbar Spine MRI using a Combination of Deep Learning Feature and Machine Learning Classifier.PLOS ONE

Dear Dr. Sudirman,

Thank you for submitting your manuscript to PLOS ONE. After careful consideration, we feel that it has merit but does not fully meet PLOS ONE’s publication criteria as it currently stands. Therefore, we invite you to submit a revised version of the manuscript that addresses the points raised during the review process.

We look forward to receiving your revised manuscript.

Kind regards,

Khanh N.Q. Le

Academic Editor

PLOS ONE

Journal Requirements:

Reviewers' comments:

Reviewer's Responses to Questions

**Comments to the Author**

1. If the authors have adequately addressed your comments raised in a previous round of review and you feel that this manuscript is now acceptable for publication, you may indicate that here to bypass the “Comments to the Author” section, enter your conflict of interest statement in the “Confidential to Editor” section, and submit your "Accept" recommendation.

Reviewer #2: All comments have been addressed

2. Is the manuscript technically sound, and do the data support the conclusions?

Reviewer #2: Partly

3. Has the statistical analysis been performed appropriately and rigorously? 

Reviewer #2: No

4. Have the authors made all data underlying the findings in their manuscript fully available?

Reviewer #2: No

5. Is the manuscript presented in an intelligible fashion and written in standard English?

Reviewer #2: Yes

6. Review Comments to the Author

Reviewer #2: 1. A large number of deep learning models have been applied to experiments, but their training details have been ignored.

2. Many figures are still of low quality and cannot be seen clearly.

3. Intermediate experimental results should also be presented.

7. PLOS authors have the option to publish the peer review history of their article (what does this mean?). If published, this will include your full peer review and any attached files.

Reviewer #2: No

---

## [Author Response · Author response to Decision Letter 2]

18 Nov 2021

Please see the attached document entitled "Response to Reviewer.docx" which includes the text below and relevant images.

Reviewer #2: 

1. If the authors have adequately addressed your comments raised in a previous round of review and you feel that this manuscript is now acceptable for publication, you may indicate that here to bypass the “Comments to the Author” section, enter your conflict of interest statement in the “Confidential to Editor” section, and submit your "Accept" recommendation.

Reviewer #2: All comments have been addressed

2. Is the manuscript technically sound, and do the data support the conclusions?

Reviewer #2: Partly

3. Has the statistical analysis been performed appropriately and rigorously?

Reviewer #2: No

We believe we have included comprehensively the experimental results in the manuscript. We implemented each of the 759 method combinations 20 times, each with a different combination of training and test sets, to provide us with statistically representative results. And we shown the average result of the 20 repeats of each method combination using four performance metrics. 

4. Have the authors made all data underlying the findings in their manuscript fully available?

Reviewer #2: No

We had made our raw MRI data, PNG images, MATLAB and PYTHON source code, and experimental results available in all previous submissions through Mendeley Data (link provided in the Data Statement). This section, if we recall correctly, has not been commented as NO before. We believe that there may be a confusion on how to download the file since in order to access the file one has to have an Elsevier account. Creating this account is free. So, we added more information on the Data Statement document regarding this.

5. Is the manuscript presented in an intelligible fashion and written in standard English?

Reviewer #2: Yes

6. Review Comments to the Author

Reviewer #2: 

1. A large number of deep learning models have been applied to experiments, but their training details have been ignored.

The training details of each model were given in Table 4 and Table 5. The tables contain both the fixed settings (the ones that do not get searched by the hyperparameter optimization process) and the variable settings/parameters (the ones that are searched automatically by the hyperparameter optimization process).

But we have rechecked the information in the table for completeness to make sure we have included all the training parameters and settings. As a result, we have added three additional parameters to Table 5 (for Fully Connected Neural Network training).

2. Many figures are still of low quality and cannot be seen clearly.

We have made queries to PLOS ONE regarding this since we have followed all the required steps to produce the figures. The original figures that we uploaded are of highest quality and very clear but for some reason when they are embedded to the pdf their quality is reduced significantly. However, the original image file can be accessed by clicking the link at the top-right of the page where the figure is displayed (see the illustration below).

3. Intermediate experimental results should also be presented.

Unfortunately, we cannot make this change since we are not sure what intermediate experimental results are needed. We have tried searching the literature for what intermediate experimental results might be, but we cannot find anything. We have also reached out to the PLOS ONE editorial team requesting for more description from the reviewers. Unfortunately, the team did not receive any response from the reviewers by the resubmission deadline. However, it is our we firm belief that our paper has included all the necessary results to base our findings. Furthermore, since we made the data and source code available, the readers should be able to reproduce the experimental results including any intermediate results if so desired.

---

## [Editor Report · Decision Letter 3]

9 Dec 2021

Automated Selection of Mid-Height Intervertebral Disc Slice in Traverse Lumbar Spine MRI using a Combination of Deep Learning Feature and Machine Learning Classifier.

PONE-D-21-04349R3

Dear Dr. Sudirman,

We’re pleased to inform you that your manuscript has been judged scientifically suitable for publication and will be formally accepted for publication once it meets all outstanding technical requirements.

Kind regards,

Nguyen Quoc Khanh Le

Academic Editor

PLOS ONE
---

## [Editor Report · Acceptance letter]

5 Jan 2022

PONE-D-21-04349R3 

Automated Selection of Mid-Height Intervertebral Disc Slice in Traverse Lumbar Spine MRI using a Combination of Deep Learning Feature and Machine Learning Classifier. 

Dear Dr. Sudirman:

I'm pleased to inform you that your manuscript has been deemed suitable for publication in PLOS ONE. Congratulations! Your manuscript is now with our production department. 

Kind regards, 

on behalf of

Dr. Nguyen Quoc Khanh Le 

Academic Editor

PLOS ONE